# Protective anti-prion antibodies in human immunoglobulin repertoires

Assunta Senatore[1] ⓘ, Karl Frontzek[1,†] ⓘ, Marc Emmenegger[1,†] ⓘ, Andra Chincisan[1], Marco Losa[1] ⓘ,
Regina Reimann[1] ⓘ, Geraldine Horny[2], Jingjing Guo[1] ⓘ, Sylvie Fels[2] ⓘ, Silvia Sorce[1], Caihong Zhu[1],
Nathalie George[2] ⓘ, Stefan Ewert[2], Thomas Pietzonka[2] ⓘ, Simone Hornemann[1,**] ⓘ &
Adriano Aguzzi[1,*] ⓘ

## Abstract

Prion immunotherapy may hold great potential, but antibodies against certain PrP epitopes can be neurotoxic. Here, we identified > 6,000 PrP-binding antibodies in a synthetic human Fab phage display library, 49 of which we characterized in detail. Antibodies directed against the flexible tail of PrP conferred neuroprotection against infectious prions. We then mined published repertoires of circulating B cells from healthy humans and found antibodies similar to the protective phage-derived antibodies. When expressed recombinantly, these antibodies exhibited anti-PrP reactivity. Furthermore, we surveyed 48,718 samples from 37,894 hospital patients for the presence of anti-PrP IgGs and found 21 high-titer individuals. The clinical files of these individuals did not reveal any enrichment of specific pathologies, suggesting that anti-PrP autoimmunity is innocuous. The existence of anti-prion antibodies in unbiased human immunological repertoires suggests that they might clear nascent prions early in life. Combined with the reported lack of such antibodies in carriers of disease-associated *PRNP* mutations, this suggests a link to the low incidence of spontaneous prion diseases in human populations.

**Keywords** anti-PrP antibodies; human immunological repertoires; next-generation sequencing; phage display; prion disease
**Subject Categories** Immunology; Neuroscience

## Introduction

Many neurodegenerative syndromes, including prion diseases, Alzheimer's disease, and Parkinson's disease, go along with the accumulation of misfolded and aggregated proteins in the central nervous system. Antibodies against such proteins may be beneficial

(Schenk *et al*, 1999), e.g., by opsonizing pathological aggregates and mediating their degradation by phagocytic cells (Heppner *et al*, 2001; Kranich *et al*, 2010). While the clinical effectiveness of antibody-based therapies against neurodegenerative diseases is still being debated (Schilling *et al*, 2018), there is ample evidence that both active immunization and passive antibody transfer can effectively clear pathological aggregates in preclinical animal models and, to some extent, in affected humans.

According to the protein-only hypothesis, the prion is an infectious particle consisting of PrP$^{Sc}$, an aggregated and proteinase-K (PK)-resistant isoform, of the cellular prion protein PrP$^C$ (Prusiner, 1998). PrP$^C$ consists of a C-terminal globular domain (GD) and an N-terminal flexible tail (FT) which includes the octapeptide repeat (OR) region, two cationic charge clusters (CC1 and CC2) and a hydrophobic core (HC; Riek *et al*, 1997). The CC1 domain of PrP$^C$ participates in Schwann cell maintenance by activating the G protein-coupled receptor Adgrg6 (Bremer *et al*, 2010; Kuffer *et al*, 2016). While PrP$^C$-deficient mice are only mildly affected, PrP$^{Sc}$ necessitates PrP$^C$ for its propagation (Weissmann, 1991) and prion toxicity is transduced by PrP$^C$ onto target cells (Mallucci *et al*, 2003; Sonati *et al*, 2013). Therefore, suppression of PrP$^C$ by means of anti-PrP$^C$ antibodies represents a rational strategy against prion diseases.

Here, we panned a synthetic human antibody phage display library to explore the presence of PrP-binding antibody fragments (Fabs; Frenzel *et al*, 2016). To identify rare antibodies to poorly antigenic epitopes that may be overlooked by conventional screening technologies, we performed "next-generation" sequencing (NGS) of panning outputs after phage selections (Ravn *et al*, 2010). Several anti-PrP binders were identified and found to antagonize prion toxicity. What is more, mining of published human antibody repertoires identified sequences similar to an anti-prion phage-derived Fab which, when expressed, acted as functional PrP binders. Lastly, the interrogation of a large unselected hospital cohort ($n = 37,894$) highlighted individuals with high-titer anti-PrP autoreactivity whose clinical presentation was heterogeneous, yet unrelated to known features

---

1   Institute of Neuropathology, University of Zurich, Zurich, Switzerland
2   Novartis Institutes for BioMedical Research, Basel, Switzerland
    *Corresponding author. Tel: +41 1 255 2107; Fax: +41 1 255 4402; E-mail: adriano.aguzzi@usz.ch
    **Corresponding author. Tel: +41 1 255 2107; E-mail: simone.hornemann@usz.ch
    †These authors contributed equally to this work

of prion diseases. Therefore, anti-prion immunity can exist in human communities and is seemingly innocuous.

# Results

## Phage display selection strategy for anti-PrP Fabs

We used three rounds of phage display to screen two synthetic human Fab phage display libraries (Fig EV1A) with short (8–10 aa) and long (12–20 aa) heavy-chain complementarity-determining regions 3 (HCDR3). These libraries were constructed to mimic human antibody repertoires by combining frameworks from human germline sequences with diversified HCDR3 whose design approximated the natural gene sequences in human repertoires as compiled in the IMGT database (Giudicelli *et al*, 2006). The first and second biopanning rounds were performed against full-length recombinant mouse PrP (recPrP$_{23-231}$) to enrich for Fabs covering a large variety of PrP epitopes. To further select Fabs recognizing specific PrP epitopes, a third panning round was conducted against several different antigens in parallel, including recPrP$_{23-231}$, recPrP$_{23-110}$ spanning the FT, the GD (recPrP$_{90-231}$ and recPrP$_{121-231}$) and synthetic peptides representing CC1$_{23-50}$ spanning the CC1, N-OR$_{39-66}$ and F-OR$_{51-91}$, containing the OR, and CC2-HC$_{92-120}$, spanning CC2 and the HC.

Only few existing antibodies bind to the natively unstructured CC1$_{23-50}$ (Didonna *et al*, 2015). To optimize our chances to identify CC1$_{23-50}$ binders, and to avoid misfolding artifacts caused by nonspecific plate adsorption, the selection against the biotinylated CC1$_{23-50}$ peptide was conducted in solid phase and by liquid-phase panning followed by neutravidin (Neu)-mediated capture. Liquid-phase panning was also performed for other biotinylated peptides (N-OR$_{39-66}$, F-OR$_{51-91}$ and CC2-HC$_{92-120}$). Furthermore, we performed panning rounds using a matrix of stringent washing conditions (Appendix Table S1) to also include an affinity readout to the analysis of the NGS screening.

## Next-Generation Sequencing (NGS) of anti-PrP Fabs

The HCDR3 domains can contribute crucially to antigen binding (Xu & Davis, 2000). Sequencing of the outputs of the third panning rounds yielded 4,847 and 11,948 unique HCDR3 sequences in 100,000 analyzed sequences for the short and long HCDR3 Fab libraries, respectively. We excluded all HCDR3 that had any counts in the negative-control outputs (Neu and BSA panning) and retained only clones with ≥ 1 read in the recPrP$_{23-231}$ output. These constraints reduced the unique HCDR3 sequences to 1,173 and 4,832 anti-PrP Fabs, respectively. We then compared the read counts of each HCDR3 between panning to recPrP$_{23-231}$ and the different PrP domains. For each HCDR3, we considered the enrichment of NGS counts in a panning output as reflecting the binding to the respective PrP peptide/fragment used in the panning. As an example of the stringent sorting criteria for epitope binding profile determination (Appendix Table S2), all HCDR3 having counts > 0 in CC1$_{23-50}$ and in recPrP$_{23-110}$ outputs, and count = 0 in all other panning outputs, were classified as specific PrP binders in the CC1$_{23-50}$ region. HCDR3 sequences were clustered based on their NGS-binding profile and found to represent a highly diverse collection of anti-PrP Fabs (Fig EV1A and Fig 1A). Predominant clones binding

to the high antigenic epitopes, defined as those with ≥ 20 NGS counts in recPrP$_{23-231}$ panning, were mostly directed against the CC2-HC$_{92-120}$ in both HCDR3 libraries (Fig 1B). Rare clones against less antigenic epitopes, i.e. displaying only one count in recPrP$_{23-231}$ panning, were predominantly showing an NGS-binding profile to the CC1$_{23-50}$ and the GD domains (Fig 1B).

We retrieved clones of interest, as identified by the HCDR3 read profile in NGS, by overlapping PCR from the third-round polyclonal phagemid DNA (Fig EV1B–E). In one instance, we designed primers to an HCDR3 sequence with an NGS-binding profile to the CC2-HC$_{92-120}$ epitope (NGS read enrichment in the CC2-HC$_{92-120}$ panning as compared to reads in panning to other PrP domains, Fig EV1B) and retrieved entire Fab sequences by PCR (Fig EV1C). The retrieved Fab, designated as FabRTV, was cloned into an *E. coli* expression vector, Sanger-sequenced, and purified by immobilized metal ion affinity chromatography (IMAC; Fig EV1D). Enzyme-linked immunosorbent assay (ELISA) confirmed the NGS-binding profile of FabRTV to the CC2-HC$_{92-120}$ domain (Fig EV1B and E).

## Identification of anti-PrP Fabs by ELISA screening

As a complementary approach, we screened 4416 clones (randomly selected by an automated colony picker) by ELISA. We then selected 312 hits reactive to recPrP$_{23-231}$, to recPrP fragments, or to synthetic biotinylated peptides (> 10 or > 5-fold over background, while displaying a Neu signal < 5-fold over background). From those, eighty confirmed anti-PrP Fabs were Sanger-sequenced, produced in *E. coli*, and epitope-mapped by ELISA (Fig EV2). Out of 49 Fabs, four targeted CC1$_{23-50}$, 15 OR$_{51-91}$, 22 CC2-HC$_{92-120}$, and eight the GD (Dataset EV1). The abundance of CC2-HC$_{92-120}$ targeting Fabs is in agreement with the NGS analysis showing that CC2-HC$_{92120}$ binders are the predominant clones. For 35 of these Fabs, the ELISA binding results confirmed the epitope binding profile determined by NGS analysis.

## Kinetic measurements by surface plasmon resonance (SPR) and affinity maturation

The K$_D$s (determined by SPR as k$_{off}$/k$_{on}$ ratio) of all tested Fabs to recPrP$_{23-231}$ were in the range of 10–100 nM (Table 1) with fast dissociation rate constants ($k_{off} > 10^{-3}$ [s$^{-1}$]).

To optimize the binding properties of Fab3 and Fab71, to CC1$_{23-50}$ and OR$_{51-91}$, respectively, we used affinity maturation libraries in which the parental HCDR2 and LCDR3 loops were replaced with pre-built highly diversified cassettes. In addition, phage display selections were repeated with more stringent conditions than in the original selection. To ensure retention of specificity to the respective epitopes, panning against recPrP$_{23-231}$ was alternated with panning to the CC1$_{23-50}$ fragment for Fab3 and to the OR$_{51-91}$ fragment for Fab71. Again, selected Fabs were screened by NGS and ELISA. After high-throughput off-rate and on-rate ELISA screening, 19 and five affinity-matured versions of Fab3 (Fab81-Fab99) and Fab71 (Fab100-Fab104) were identified, respectively. These clones were also prevalent in the NGS dataset. With this strategy, the EC$_{50}$s of the affinity-matured Fabs were improved by 2–3 logs over the parental Fabs (Fig 2A and Appendix Table S3).

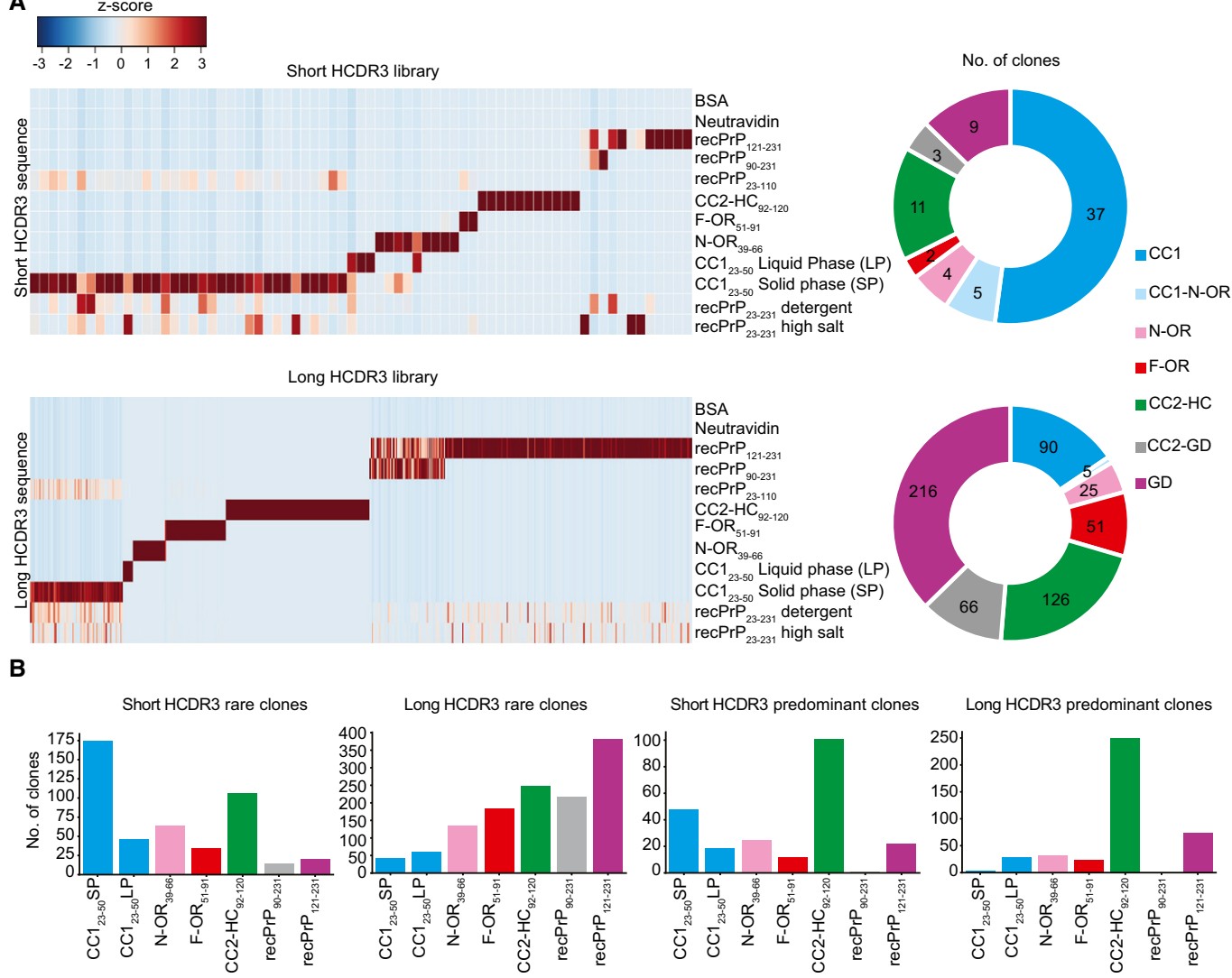

**Figure 1. Identification of PrP binders.**

A   Heat maps representing enriched HCDR3 sequences across the different panning sets for the short- and long-HCDR3 (upper and lower panels, respectively) phagemid libraries. HCDR3 sequences were selected based on NGS counts in 100,000 analyzed sequences (Z-score values; Appendix Table S2) and clustered according to the NGS-binding profiles. Red and blue: high and low number of NGS counts of the HCDR3 sequence, respectively. Donut charts (the right side) of each heat map indicate the number of clones with NGS-identified HCDR3 for a predicted PrP epitope.

B   Bar graphs showing the number of rare (one count in mouse recPrP$_{23-231}$ panning) and predominant (count $\geqq$ 20) clones binding to the various PrP regions for the short- and long-HCDR3 libraries.

## Epitope confirmation and mapping

Next, we confirmed the binding behavior of the different Fabs to biotinylated FT-PrP peptides CC1$_{23-50}$, N-OR$_{39-66}$, F-OR$_{51-91}$ and CC2-HC$_{92-120}$ by SPR (Appendix Fig S1). For all tested Fabs, the specificity of the targeted epitope matched the ELISA epitope profiling. Being unstructured, the FT displays many linear epitopes (Polymenidou *et al*, 2008). Therefore, we additionally mapped the epitopes of the anti-FT Fabs by competition ELISA using overlapping dodecameric PrP peptides (each shifted by two residues) spanning residues 23–120. The binding of Fab3 and of its affinity-matured version Fab83 to immobilized recPrP$_{23-231}$ was blocked

by peptides P1 and P2 (residues 23–34 and 25–36, respectively; Fig 2B and C), pointing to the KKRPKPG polybasic stretch as their minimal epitope.

The anti-OR$_{51-91}$ Fab8, Fab12, Fab71 and the affinity-matured Fab100 recognized the sequence, WGQPHGGG(S)WGQ, which is repeated four times in PrP (residues 55–66, 64–74, 72–82, and 80–90). Additionally, Fab8, Fab12, and Fab71 also targeted NRYPPQGGTWGQ at the very beginning of the OR domain (residues 47–58; Fig 2D and E and Appendix Fig S2). Fab44 recognized the sequence WGQPHGG within the OR (Appendix Fig S2). For Fab7, Fab10, Fab41, and Fab52, we could not identify any PrP peptide that abrogated the ELISA signal to recPrP$_{23-231}$. This may point to the

**Table 1.** Dissociation rate ($k_{off}$) and equilibrium (KD) constants of the PrP-binding Fabs determined by SPR

| | $k_{off}$ (1/s) | KD (M) |
|---|---|---|
| Fab_POM1 | 2.96E-04 | 2.01E-09 |
| Fab1 | 2.88E-03 | 1.21E-08 |
| Fab2 | 1.58E-03 | 1.01E-08 |
| Fab3 | 2.46E-03 | 7.72E-09 |
| Fab4 | 1.93E-03 | 1.41E-08 |
| Fab6 | 1.44E-03 | 9.89E-09 |
| Fab7 | 2.41E-03 | 1.76E-08 |
| Fab8 | 3.62E-03 | 1.79E-08 |
| Fab10 | 2.61E-03 | 1.41E-08 |
| Fab12 | 3.20E-03 | 1.71E-08 |
| Fab13 | 4.98E-03 | 4.56E-08 |
| Fab15 | 3.76E-03 | 1.27E-08 |
| Fab25 | 2.42E-03 | 4.71E-08 |
| Fab28 | 1.08E-03 | 1.28E-08 |
| Fab29 | 3.15E-03 | 1.04E-08 |
| Fab30 | 1.76E-03 | 4.76E-09 |
| Fab32 | 2.90E-03 | 1.50E-08 |
| Fab35 | 2.29E-03 | 1.75E-08 |
| Fab41 | 2.65E-03 | 1.95E-08 |
| Fab44 | 3.18E-03 | 1.21E-08 |
| Fab46 | 3.39E-03 | 1.46E-08 |
| Fab48 | 2.45E-03 | 1.06E-08 |
| Fab52 | 4.12E-03 | 2.98E-08 |
| Fab53 | 1.73E-03 | 9.79E-09 |
| Fab61 | 3.97E-03 | 2.06E-08 |
| Fab69 | 1.36E-02 | 9.50E-08 |
| Fab71 | 2.45E-03 | 6.20E-09 |
| Fab72 | 2.14E-03 | 7.14E-09 |
| Fab74 | 1.57E-02 | 4.85E-08 |
| Fab75 | 3.67E-03 | 2.64E-08 |

presence of conformational/discontinuous epitopes. The CC2-HC$_{92-120}$ directed Fab13, Fab53, Fab61, and Fab69 all recognized residues 93–100 (GTHNQWNK; Appendix Fig S2).

In addition, Fab83, Fab100, Fab53, and Fab74 were also able to detect wild-type PrP$^C$ (wtPrP$^C$) in brain homogenates (BH) from wild-type (C57BL6/J) and tga20 mice (Fischer et al, 1996) overexpressing wtPrP$^C$, but not in the brains of mice expressing PrP deletion mutants lacking the respective epitopes (Fig EV3B).

The CC1$_{23-50}$ binder Fab83 did not detect PrPΔ25–50 (Yoshikawa et al, 2008) and PrPΔ32–93 (Shmerling et al, 1998) on Western blots, nor did it detect cell expressed PrPΔ23–27, PrPΔ23–31 and PrP mutated in the polybasic stretch from KKRPK to KPRKK by ELISA (Fig EV3C). Lysine-to-alanine substitutions within the KKRPK sequences reduced the binding of Fab83. Hence, the Fab83 epitope comprises both lysine K24 and proline P26 in the N-terminal KKRPK stretch (PrP23–27) and additional residues in the PrP32–36 segment.

Alternatively, KKRPK could have become buried in the truncated version PrPΔ32–93.

Consistent with the peptide epitope mapping, Fab100 did not react with the OR-deleted PrPΔ32–90 and PrPΔ51–90 (Yoshikawa et al, 2008). In addition, Fab53 did not detect PrPΔ94–110 (Bremer et al, 2010), whereas the anti-GD Fab74 recognized all FT-modified versions of PrP. Most of the tested Fabs also cross-reacted with human recPrP$_{23-230}$ (Fig EV3A).

We then performed immunoprecipitations of wtPrP$^C$ from BH of wt mice and from mice expressing different PrP deletion mutants, with Fab83 (CC1$_{23-50}$) and Fab71 (OR$_{51-91}$; Fig EV4A and B). Fab83 and Fab71 immunoprecipitated wtPrP$^C$ but not PrPΔ25–50 and PrPΔOR with deletion of the Fab binding sites as predicted from the ELISA. wtPrP$^C$ was eluted by the P1 peptide containing the Fab83 binding site within the CC1$_{23-50}$ but not by the P15 peptide within the OR$_{51-91}$ (Fig EV4A). Similarly, Fab71 immunoprecipitated wtPrP$^C$ was eluted from the Fab71-beads complex by the epitope-mimicking peptide P17 in the OR$_{51-91}$, but not by the P3 peptide within the CC1$_{23-50}$ (Fig EV4B).

We then tested Fab100 for immunoprecipitation of PrP$^{Sc}$ from prion infected brain homogenates (Fig EV4C). Fab100 immunoprecipitated PrP$^C$ and PrP$^{Sc}$ from NBH and prion-infected brains, respectively. Elution under native conditions was achieved using peptide P17 (within OR$_{51-91}$) but not P2 (within CC1$_{23-50}$), confirming the specificity of Fab100 for the OR of both PrP$^C$ and PrP$^{Sc}$. Proteinase-K (PK) digestion assays confirmed the presence of PrP$^{Sc}$ in the eluted fractions from prion-infected BH after immunoprecipitation by Fab100 (Fig EV4D).

### Validation by flow cytometry and immunostaining

We also assessed the binding of a panel of Fabs to cell-surface wtPrP$^C$ by flow cytometry using the murine neuroblastoma cell line CAD5. For each Fab, we compared the mean fluorescence intensity (MFI) between CAD5 $Prnp^{+/+}$ and $Prnp^{-/-}$ cells (Appendix Fig S3A and B). All CC1$_{23-50}$ binders, the majority of OR$_{51-91}$ binders and CC2-HC$_{92-120}$ binders discriminated between $Prnp^{+/+}$ and $Prnp^{-/-}$ CAD5 cells, with Fab71 being the best performer. Of the GD binders, only Fab74 showed a differential signal, suggesting that the GD of membrane-bound wtPrP$^C$ is less accessible to antibodies than the FT. Binders were tested by immunofluorescence on paraffin-embedded brain sections of C57BL6/J, $Prnp^{ZH3/ZH3}$ and tga20 mice. Fab3, Fab71, and Fab74 detected murine wtPrP$^C$ (Appendix Fig S3C) with high specificity.

### Activity of Fabs in models of prion disease

Antibodies that bind to the FT have been demonstrated to be neuroprotective, whereas those directed against certain epitopes of the GD are invariably toxic (Sonati et al, 2013; Reimann et al, 2016). We therefore asked whether Fabs against CC1$_{23-50}$ (Fab3 and Fab83), the OR$_{51-91}$ (Fab8, Fab44, Fab71, and Fab100), and the GD (Fab25, Fab1, and Fab29) may counteract neurotoxicity in prion-infected cerebellar slices (COCS; Sonati et al, 2013; Herrmann et al, 2015). Tga20 COCS were exposed to Rocky Mountain Laboratory prions (passage #6, RML6) or to NBH and cultured in the presence or absence of the different Fabs (550 nM). At 45 days post-infection (dpi), prion-infected slices showed

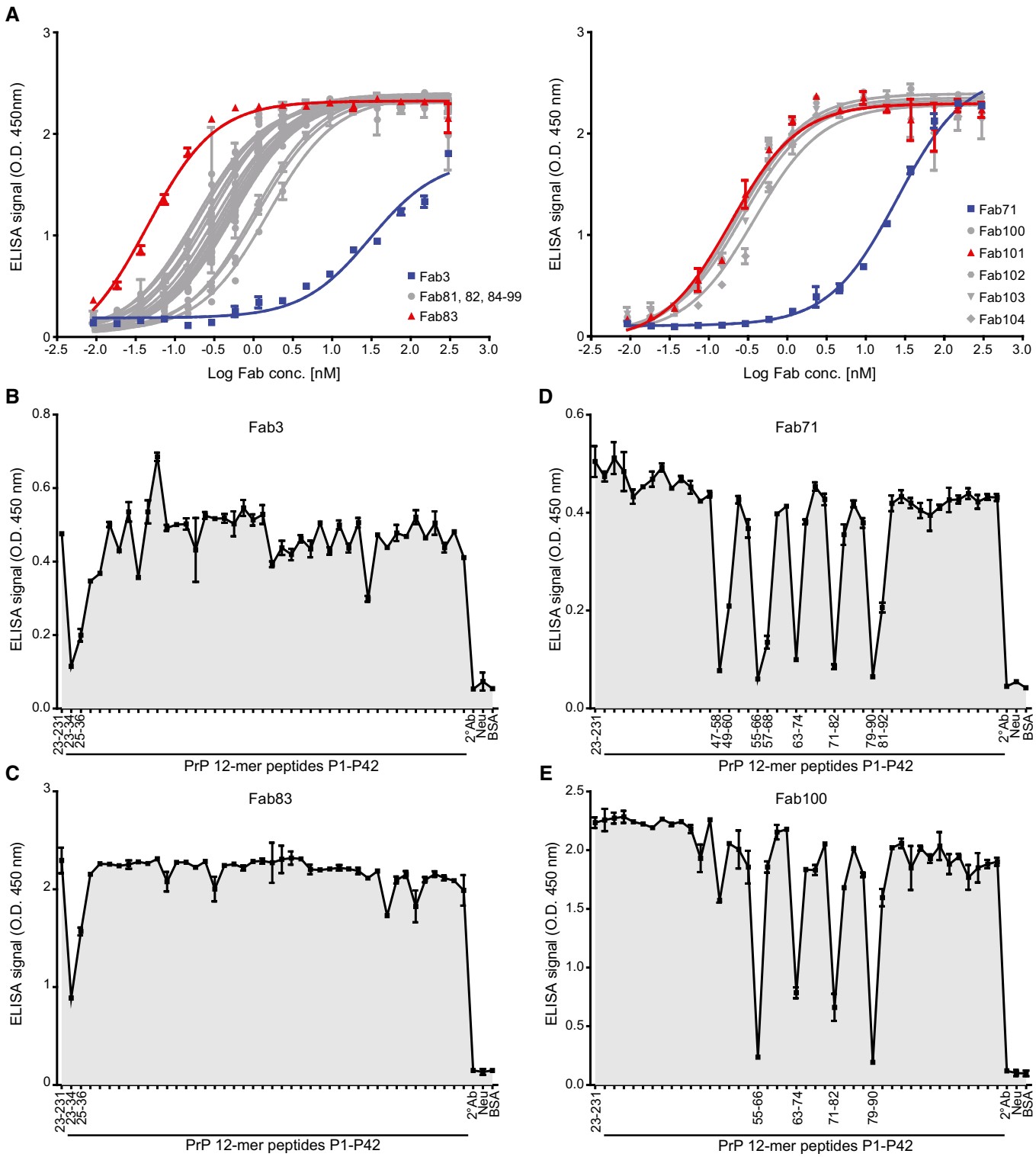

**Figure 2. Binding specificities of retrieved anti-PrP Fabs by ELISA.**

A    ELISA titration curves (OD at 450 nm) of Fab3 (blue; left panel) and Fab71 (blue; right panel) against mouse recPrP$_{23-231}$ compared to their best affinity-matured versions, Fab83 and Fab101 (red; EC$_{50}$ values in Dataset EV1). Curves were fitted by non-linear least-squares regression analysis for statistical analysis: log(agonist) vs. response.

B–E   FT-peptide competition ELISA to map the epitopes of the indicated Fabs. Peptides that strongly inhibit the binding of the Fabs to recPrP$_{23-231}$ are indicated by their residue numbers in the PrP sequence and reflect the respective binding epitopes. Positive control: recPrP$_{23-231}$; negative controls: Neu, BSA, and the secondary antibody (2° Ab).

Data information: ELISA data were performed in duplicates. Data represent the mean ± sem.

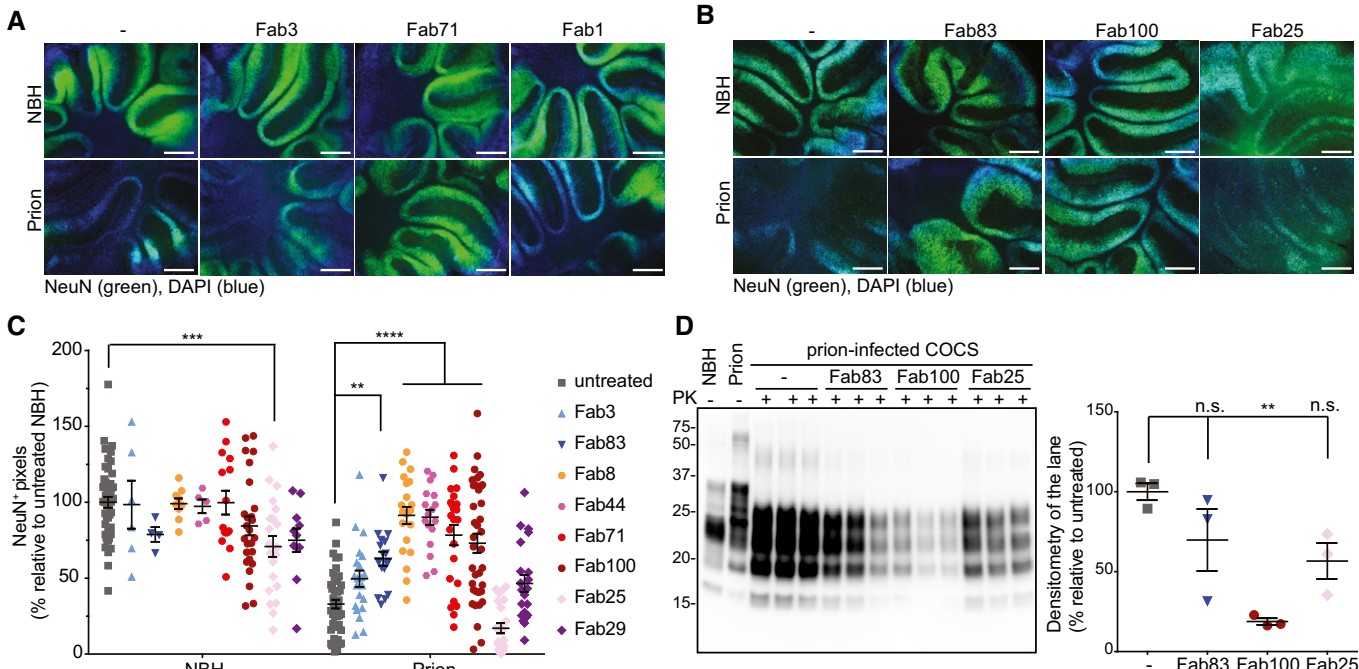

**Figure 3.** **Fabs targeting the CC1 and OR of PrP$^C$ prevent prion-induced neurotoxicity in COCS.**

A   Fluorescence micrographs of *tga20* slices chronically exposed to prions and cultured in the presence of either Fab3 (CC1$_{23-50}$), Fab71 (OR$_{51-91}$), or Fab1 (GD) for 45 days. Scale bar: 500 μm.

B   Same as (A), but in the presence of the affinity-matured antibodies.

C   NeuN immunofluorescence coverage of prion-infected slices cultured in the presence of the Fabs. Antibodies targeting the CC1$_{23-50}$ and the OR$_{51-91}$ afforded protection against neurodegeneration. NBH: non-infectious brain homogenate. Each dot represents a cerebellar slice for all treatment groups. Data are indicated as the mean ± sem. Two-way ANOVA followed by Bonferroni's *post hoc* test; **$P < 0.01$; ***$P < 0.001$; ****$P < 0.0001$.

D   Western blot analysis of COCS lysates. Prion-infected slices treated with Fab100, but not with Fab83 and Fab25 showed reduced levels of PrP$^{Sc}$. $n = 3$ biological replicates. Data are indicated as mean ± sem. One-way ANOVA followed by Bonferroni's *post hoc* test was used for statistical analysis: **$P < 0.01$, n.s.: not significant.

conspicuous neurodegeneration (Fig 3A–C). The GD binder Fab25 showed intrinsic neurotoxicity as indicated by loss of NeuN immunoreactivity in NBH-treated COCS. All tested OR$_{51-91}$ binders, but none of the GD binders prevented prion-induced neurotoxicity. Fab3 (binding to CC1$_{23-50}$) did not prevent prion neurotoxicity, whereas its affinity-matured derivative Fab83 was effective (Fig 3A–C).

We then investigated the effects of the Fabs on prion replication and PrP$^{Sc}$ accumulation (Fig 3D). Treatment with the anti-OR$_{51-91}$ Fab100, but not by anti-CC1$_{23-50}$ Fab83, reduced PrP$^{Sc}$ in prion-infected COCS, although both were neuroprotective. In slices treated with the GD Fab25, PrP$^{Sc}$ levels were not significantly different from untreated prion-inoculated slices.

We also assessed the ability of the Fabs to arrest prion replication and/or PrP$^{Sc}$ accumulation in prion susceptible cells. CAD5 *Prnp$^{+/+}$* and CAD5 *Prnp$^{-/-}$* cells were exposed to prions or NBH and treated with 10 μg/ml of either CC1$_{23-50}$ binder Fab3, OR$_{51-91}$ binder Fab71 or the GD targeting Fab29. Fabs were added to the medium 1 h after infection and after every splitting. At 14 dpi, PrP$^{Sc}$ content was determined using a homogenous-phase PrP$^{Sc}$ time-resolved (TR) FRET quantification immunoassay after proteolytic PrP$^C$ removal. PrP$^{Sc}$ was seen by TR-FRET in prion-infected CAD5 *Prnp$^{+/+}$* but not in CAD5 *Prnp$^{-/-}$* cells indicating that the initial prion inoculum had been diluted below detectability. Fab83 did not reduce PrP$^{Sc}$ levels in

CAD5 *Prnp$^{+/+}$* cells despite being protective in COCS (Fig EV5B). Fab71 and Fab100, but not Fab3 and Fab29, lowered PrP$^{Sc}$ in prion-infected CAD5 *Prnp$^{+/+}$* cells (Fig EV5A and B). Hence, OR$_{51-91}$ binders might exert their neuroprotective activity by reducing prion accumulation.

## Identification of anti-PrP antibodies from human antibody repertoires similar to the phage-derived Fabs

The Fabs were derived from a synthetic library mimicking the human antibody repertoire, raising the question whether analogous antibodies might be present in bona fide human repertoires. The heavy-chain and light-chain frameworks of Fab71 correspond to human germlines VH3–30 and Vκ3–15, respectively, whereas its HCDR3 originates from the library randomization strategy. We therefore searched for human antibodies harboring HCDR3 amino acid sequences similar to that of Fab71. We examined large-scale repertoire datasets with billions of sequences that were generated recently by sequencing heavy-chain (VH; here referred to as DW (DeWitt *et al*, 2016) and BB (Briney *et al*, 2019) datasets) and natively paired heavy/light-chain variable regions (VH:VL; DK dataset; DeKosky *et al*, 2016) from circulating naïve and memory B cells of healthy donors. While no exact match to Fab71 HCDR3 was found, twelve out of fourteen analyzed donors displayed 629 HCDR3 sequences that differed from

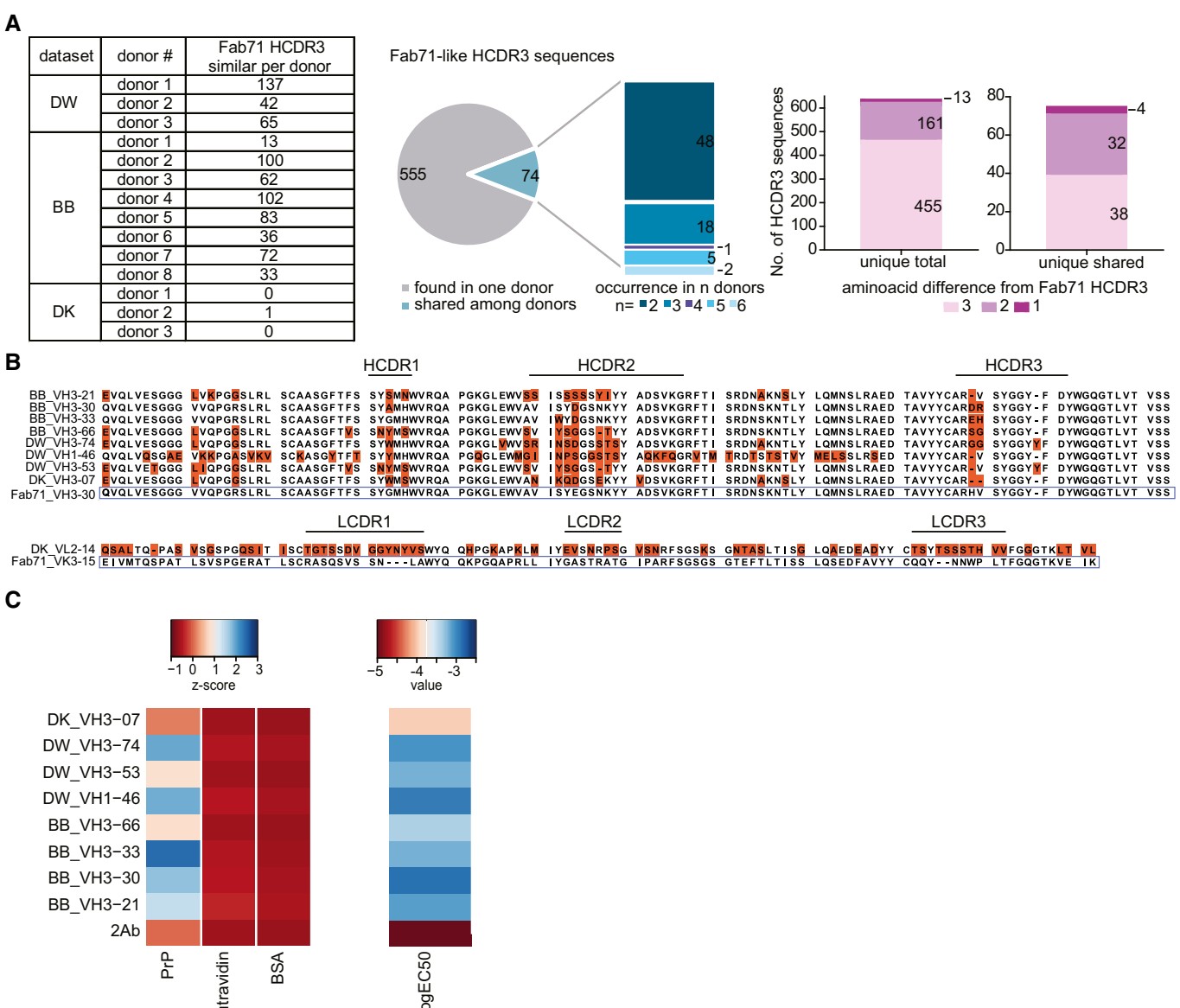

**Figure 4. Identification of anti-PrP Fab71 analogous antibodies in human antibody repertoires.**

A   Fab71-similar HCDR3 sequences in three different NGS datasets (DW: dataset for VH:VL; BB and DK datasets for VH) of human antibody repertoires from healthy donors. Pie chart and stacked bar plot (middle) indicate the occurrence of the identified Fab71 similar HCDR3s: 555 out of 629 identified sequences were only found in one donor, while 74 sequences were shared between donors and occurred in 2 up to six donors. Stacked bar plots (right) report the number of sequences differing to Fab71 HCDR3 by 3, 2 or 1 residues. In total, 13 sequences differed by only one residue. Among them, four sequences were shared between different donors.

B   Sequence alignment of Fab71 VH3–30 with HCDR3 regions from the different databases that differ from Fab71 by ≤ 3 residues (upper panel). Amino acids differing from Fab71 are highlighted by orange boxes. Lower panel: sequence alignment of Fab71 light-chain VK3–15 with DK_VL2–14 that is naturally paired with DK_VH3-07.

C   Left: Heat map showing the binding specificities (Z-scores) of selected Fab71 human analogous antibodies to human recPrP$_{23-230}$ compared to the negative controls (BSA and Neu). Right: Heat map representing the reactivity (−logEC50) obtained from dose-dependent ELISA binding curves of the analogous antibodies to human recPrP$_{23-230}$. Red: low reactivity; blue: high reactivity.

Fab71 HCDR3 by ≤ 3 residues (Fig 4A). Among them, 74 HCDR3 sequences occurred in ≥ 2 subjects, four of which differed from Fab71 HCDR3 by only one residue (Fig 4A).

We selected eight heavy-chain variable regions similar to Fab71. In one of these, HCDR3 deviated from Fab71 in just two residues and matched the Fab71 VH3–30 segment. Three further

HCDR3 differed from Fab71 HCDR3 by only one residue and two of those HCDR3 appeared in antibodies using different V genes (VH3–21 and VH1–46) from three subjects (Fig 4B). We expressed these heavy-chain variable regions in bacteria along with the Fab71 light chain to purify Fabs and test their reactivity to human recPrP$_{23-230}$. In the case of DK_VH3-07, where VL

sequence information was available from the VH:VL paired sequencing dataset, we used VL_2-14. All selected naturally occurring Fab71-like antibodies, except the DK_VH3-07, recognized human recPrP$_{23-230}$ by ELISA specifically (Fig 4C). BB_VH3-

30, DW_VH1-46 and DW_VH3-74 Fabs, displayed the highest apparent affinities (EC50 = 1.21 μM, EC50 = 1.07 μM, and EC50 = 0.98 μM, respectively; for DK_VH3-07 Fab, which did not bind to human recPrP$_{23-230}$, calculated EC50 = 62.8 μM). While

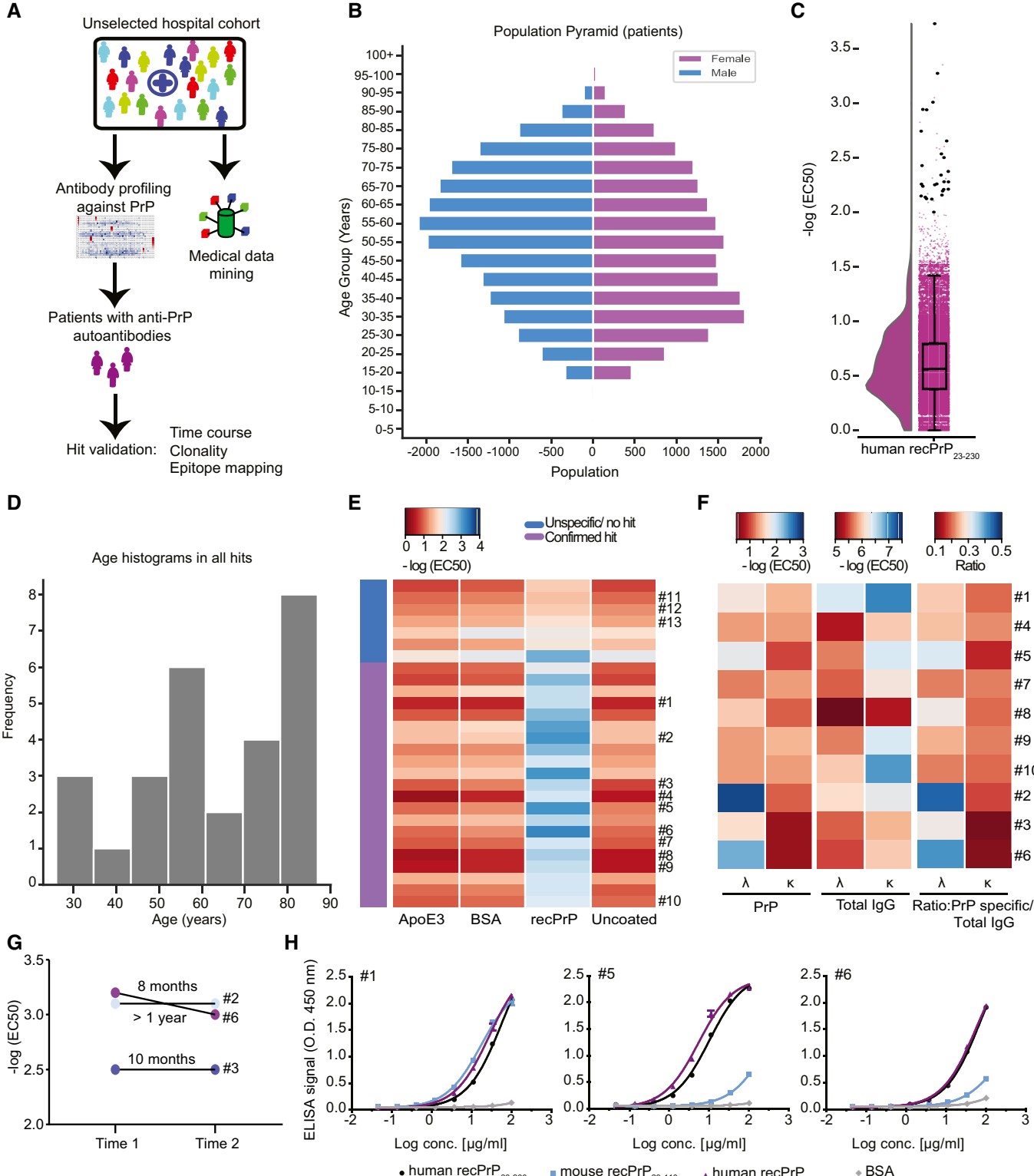

**Figure 5.**

Figure 5. Identification of anti-PrP autoantibodies in an unselected hospital patient cohort.

A  Schematic workflow for the HTS of patient plasma samples for anti-PrP autoantibodies.
B  Age pyramid, separated for females and males, of the 37,894 unselected hospital patients tested in the HTS.
C  Boxplot with half-violin plot displaying the distribution of autoantibody reactivity in 48,718 plasma samples. Twenty-seven samples over the reactivity threshold of $-\log(EC50) \geq 2$ and fitting error < 20% were considered as hits in the primary screen. Black dots correspond to single samples. The boxplot divides the dataset into three quartiles. The box extends from the first to the third quartile values of the dataset, with a line at the median. Whiskers show the range of the data (minimum and maximum).
D  Frequency distribution of hits based on age.
E  Heat map representation for the validation of the hits from the primary screen (21 out of 27 hits). Confirmed hits displayed selective reactivity against human recPrP$_{23-230}$. Controls: Patients #11–13, ApoE4 and BSA. Red: high reactivity; blue: low reactivity.
F  Heat maps representing autoantibody clonality of hits analyzed by ELISA. Autoantibodies were mostly constituted by lambda light chains (λ), while the kappa light chains (κ) showed prevalence in the total IgG fraction.
G  Determination of antibody reactivity (-log(EC50)) for three individuals at two different time points. Individuals were selected by displaying strong anti-PrP IgG reactivity in the first test (Time 1). Reactivity was maintained for even more than one year (Time 2).
H  Purified IgGs from patients with strong anti-PrP reactivity were tested against PrP variants and BSA (negative control) in dose-dependent ELISAs. Patient #5 and Patient #6 showed binding mostly to the human recPrP$_{121-230}$. Patient #1 displayed a polyreactive pattern with binding to the human recPrP$_{121-230}$ and mouse recPrP$_{23-110}$.

Data information: ELISA data were performed in duplicates. Data represent the mean ± sem.

these affinities are low, they may be greatly raised by dimeric cooperativity in bivalent antibodies.

## Identification of natural anti-PrP autoantibodies in an unselected hospital cohort

To assess the validity of the above findings, we interrogated a large cohort of human individuals for naturally occurring antibodies against PrP (Fig 5A and Appendix Fig S4A). We performed an automated microELISA to screen 48,718 plasma samples from 37,894 individuals admitted to almost all clinical departments of the University Hospital Zurich (Fig 5A and B and Appendix Fig S4A) for binding to human recPrP$_{23-230}$. We applied stringent criteria: Only plasma samples displaying log(EC50) ≥ −2 and logistic regression fitting error < 20% were considered as hits. In a primary high-throughput screen (HTS), 27 individuals (9 females and 18 males) were found to harbor IgG antibodies reacting to human recPrP$_{23-230}$ (Fig 5C and D). A validation screen confirmed PrP reactivity in the plasma samples of 21 individuals, indicative of a 0.06% prevalence of autoantibody carriers (Fig 5E). The clinical presentation of individuals with such autoantibodies was heterogeneous (Appendix Table S4), with no statistically significant enrichments found in disease codes (International Classification of Disease, ICD-10), medication reports, age, or sex (Fig 5D and Appendix Fig S4B and C and Appendix Table S5). None of the patients with anti-PrP reactivity showed signs of prion-related pathology. Repeated longitudinal sampling indicated that three individuals sustained their high anti-PrP titers over a time span of several months up to more than a year (Fig 5G), suggesting that these anti-PrP-autoantibodies are stable over time.

We then assessed the clonality of the anti-PrP-autoantibodies in a subset of PrP-reactive samples. First, we compared by ELISA the λ/κ light-chain ratio in total immunoglobulins vs. anti-PrP autoantibodies. We found that antibodies binding human recPrP$_{23-230}$ have preferential contribution of λ over κ light chains (Fig 5F). We then purified immunoglobulins from selected plasma samples, confirmed their specific reactivity to human recPrP$_{23-230}$, and assayed them for differential binding to recPrP$_{23-110}$ (FT) versus recPrP$_{121-230}$ (GD). While some patients showed immunoreactivity against the GD, antibodies in other patients targeted both the FT and the GD, suggesting a polyclonal antibody response (Fig 5H). These findings corroborate

the evidence for the existence of naturally occurring antibodies against the prion protein in humans.

# Discussion

Anti-PrP antibodies are effective in cells and mice infected with prions (Enari et al, 2001; Peretz et al, 2001) and may represent a plausible therapeutic strategy (Heppner et al, 2001; White et al, 2003). However, although certain anti-PrP antibodies afford neuroprotection to prion-infected mice, others cause extensive neuronal loss (Solforosi et al, 2004) and several antibodies to the GD on PrP result in acute on-target toxicity (Sonati et al, 2013; Reimann et al, 2016). Also, antibodies to the OR region of PrP prevent neurotoxicity triggered by GD-binding antibodies and by prions in organotypic slices (Sonati et al, 2013; Herrmann et al, 2015; Bardelli et al, 2018). Thus, the biological effect of anti-PrP antibodies crucially depends on the targeted PrP epitope. Hence, this study aimed to produce a high-resolution map of neuroprotective epitopes, with the ultimate goal of identifying effective immunotherapeutics.

The production of anti-PrP antibodies in wt animals is hindered by self-tolerance (Polymenidou et al, 2004). Therefore, anti-PrP monoclonal antibodies have been developed mainly in PrP-deficient mice (Prusiner et al, 1993; Polymenidou et al, 2008) or by phage display (Williamson et al, 1996) and target mostly immunodominant PrP epitopes within the central region and GD of PrP (Williamson et al, 1998).

Here, we screened a synthetic human antibody library by phage display which differs from previous approaches (Adamson et al, 2007; Flego et al, 2007) in several aspects. Firstly, we expanded the number of PrP antigens, instead of using a single PrP fragment for panning, with the intent to discover antibodies to all regions of PrP. Secondly, we deep-sequenced the panning outputs—a strategy that optimizes the detection of extremely rare antibody clones. Finally, we opted to bacterially express antibodies as Fabs which are typically more stable and less susceptible to dimerization than scFv (Arndt et al, 2001). This enabled us to generate anti-PrP Fabs with highly diverse specificities. In addition, this strategy yielded, besides the 49 Fabs identified by ELISA screening, hundreds of additional rare Fab hits against less antigenic epitopes of PrP.

Having produced a broadly diversified panel of anti-PrP Fabs, we assessed the correlation between their epitope and their biological activity. All $OR_{51-91}$-binding Fabs prevented neurodegeneration in prion-infected COCS. These results are consistent with previous findings (Sonati *et al*, 2013; Herrmann *et al*, 2015; Bardelli *et al*, 2018) pointing to the OR of PrP as the effector arm of neurotoxicity. In prion-infected cerebellar slices and cells, neuroprotection by $OR_{51-91}$ binders was associated with reduced levels of PK-resistant $PrP^{Sc}$. This effect has not been described for POM2 that blocked the toxic cascade elicited by prions, downstream of its replication. Other OR-targeting antibodies (Feraudet *et al*, 2005) were described to block $PrP^{C}$ internalization, thus reducing the rate of intracellular conversion to $PrP^{Sc}$. Alternatively, the engagement of the OR by Fab71 and Fab100 may prevent the interaction between $PrP^{C}$ and $PrP^{Sc}$, as suggested for prion-clearing anti-GD antibodies in susceptible cell lines (Enari *et al*, 2001; Peretz *et al*, 2001; Pankiewicz *et al*, 2006).

Fab83, which binds CC1 (23-KKRPK-27) with high affinity, also afforded neuroprotection. $CC1_{23-50}$-targeting antibodies are rare and, to our knowledge, were never assessed for neuroprotection against prions. However, it is well-established that the CC1 is required for the neurotoxicity of PrP mutants with deletion in the central domain (Westergard *et al*, 2011). Transgenic mice expressing PrPΔ23–31 or PrPΔ23–88 not only exhibit prolonged survival after prion inoculation but also accumulate less $PrP^{Sc}$ in their brains (Supattapone *et al*, 2001; Turnbaugh *et al*, 2012). Conversely, Fab83 did not reduce $PrP^{Sc}$ levels in CAD5 cells and in prion-infected COCS, yet it was neuroprotective. Furthermore, alanine substitutions of the positively charged amino acids within PrP 23-KKRPK-27 did not prevent $PrP^{Sc}$ formation in prion-infected neuroblastoma cells ScN2a (Abalos *et al*, 2008). We conclude that the antibody-mediated neutralization of the CC1 is neuroprotective by other means than arresting $PrP^{Sc}$ generation. We speculate that the CC1 blockade prevents the interaction of PrP with downstream effectors of neurotoxicity, such as the Group-I metabotropic glutamate receptors (Goniotaki *et al*, 2017), which in turn trigger neurotoxicity. The activity of Fab83 shows that toxicity and $PrP^{Sc}$ accumulation can be separated. A Fab preventing neurotoxicity might be efficacious even if it does not fully prevent prion accumulation. Accordingly, prevention of neurotoxicity arrested the progression to clinical disease in mice despite continued prion propagation (Mallucci *et al*, 2003). Although an antibody binding to the GD region of PrP is currently being evaluated in a clinical trial (Klyubin *et al*, 2014; Dyer, 2018), the neurotoxicity of Fab25 calls for caution with such reagents (Reimann *et al*, 2016).

The presence of anti-PrP antibodies in a human repertoire is surprising. Prions do not elicit antibody responses (Porter *et al*, 1973), most likely because of the negative selection of B cells that autoreact to $PrP^{C}$ whose primary amino acid sequence is identical with $PrP^{Sc}$ (Polymenidou *et al*, 2004). Immunization of wt mice with recombinant PrP did not result in antibodies to N-terminal epitopes, and most antibodies recognized recombinant PrP in ELISA but not native $PrP^{C}$ on cell membranes. We found many Fabs binding within the FT in the synthetic human antibody phage display library, most of which recognized native PrP on the cell surface. Unexpectedly, sequence comparisons identified PrP-reactive antibodies in published databases of naïve human repertoires from circulating B cells. Finally, we found high-titer PrP autoantibodies in the plasma of unselected hospitalized patients. Certain antibodies to PrP can mimic prion neurotoxicity (Sonati *et al*, 2013; Herrmann *et al*, 2015). At the time of the analysis, none of these subjects had received a diagnosis of prion disease. The clinical presentation of the patients with high-titer PrP autoantibodies was heterogeneous, and although signs of dementia were reported for two of the anti-PrP antibody carriers, we did not find any statistically significant correlation with neurological or any other disease when comparing subjects with and without anti-PrP autoantibodies. Similarly, PrP autoantibody titers present in a subset of *PRNP* mutation carriers neither correlated with the *PRNP* mutation status nor with the onset of clinical prion disease (Frontzek *et al*, 2020).

The frequency of high-titer anti-PrP antibody carriers (0.06%) is much lower than the occurrence of Fab71-like HCDR3 sequences in published human repertoires. This discrepancy could mean that most anti-PrP specificities exist in a dormant state, or are expressed as B-cell receptors, but do not produce circulating antibodies (Iype *et al*, 2019). It will be interesting to discover the triggers that may ignite antibody production and, possibly, afford protection against prions.

The evidence presented here indicates that the immunological repertoires of unselected humans can contain antibodies against $PrP^{C}$. Such antibodies might protect against prions. The rarity of individuals with high-titer anti-PrP immunity is unsurprising since $PrP^{C}$ is highly expressed in many immune cells including the developing thymus, which in mice results in an almost-insurmountable central tolerance (Polymenidou *et al*, 2004).

We suspect that it is not $PrP^{C}$ that induces adaptive immune responses against prions, but nascent $PrP^{Sc}$ instead. Accordingly, clinically silent prion generation may occasionally occur in healthy individuals (Edgeworth *et al*, 2010). Most $PrP^{Sc}$ aggregates arising de novo may result in exposure of neoepitopes and/or epitopes occluded in cell-borne $PrP^{C}$. The resulting immune response may clear these nascent prions, akin to the immune surveillance against neoplastic cells. Progressive senescence of adaptive immunity (Nikolich-Zugich, 2018), which is well-documented for a variety of infectious diseases, may explain why both sporadic and familial prion diseases flare up mostly in late life.

In conclusion, the generation of antibodies to the whole PrP epitope space provides new tools to understand the mechanism of neurodegeneration conveyed by prions. The presence of prion protein binders in human antibody repertoires and of anti-PrP reactivity in human plasma points to a potential source of immunotherapeutics against prion diseases.

# Materials and Methods

### Experimentation with human samples

All experiments and analyses involving samples from human donors were conducted with the approval of the local ethics committee (KEK-ZH-Nr. 2015-0561 and BASEC-Nr. 2018-01042) and in accordance with the provisions of the Declaration of Helsinki and the Good Clinical Practice guidelines of the International Conference on Harmonisation. Written informed patient consent was received by all individuals participating in this study.

## Animal experiments

All animal experiments were conducted in strict accordance with the Swiss Animal Protection law (*Tierschutzgesetz* and *Tierschutzverordnung*) of the *Swiss Bundesamt für Lebensmittelsicherheit und Veterinärwesen BLV*. The Animal Welfare Committee of the Canton of Zurich approved all animal protocols and experiments performed in this study (animal permits Versuchstierhaltung 123, ZH90/2013, ZH139/16).

## Antigen production

Synthetic PrP peptides $CC1_{23-50}$, $N\text{-}OR_{39-66}$, $F\text{-}OR_{51-91}$, and $CC2\text{-}HC_{92-120}$ were synthesized (EZBiolabs) with a convenient C-terminal biotin moiety, separated by a short linker. Recombinant mouse and human PrP proteins were produced as described (Lysek & Wuthrich, 2004).

## Construction of the synthetic human Fab phage library

A synthetic human Fab phagemid library (Novartis Institutes for BioMedical Research) was used for the phage display selections. A gene fragment encoding the germline framework combinations IGHV3–23 and IGKV1–39, IGHV3–23 and IGLV3–9, IGHV3–30 and IGKV3–15, IGHV3–15 and IGLV1–47 were synthesized by Invitrogen's GeneArt service in Fab format and cloned into a phagemid vector serving as the base templates. These human germlines were used as they display favorable framework combinations for a phage display library (Tiller *et al*, 2013). The phagemid vector consists of ampicillin resistance, ColE1 origin, M13 origin, and a bi-cistronic expression cassette under a lac promotor with OmpA – light chain followed by PhoA–heavy chain – Flag – 6xHis – Amber stop – truncated pIII (amino acids 231 – 406).

Only HCDR3 was diversified, and primers were designed to incorporate up to 11 amino acids at defined ratios mimicking their natural occurrences: aspartic acid, glutamic acid, arginine, histidine, serine, glycine, alanine, proline, valine, tyrosine, and tryptophan. Leucine and phenylalanine were also allowed at a certain position of HCDR3. Certain residues were omitted on purpose to remove potential post-translational modification sites. Randomized primer synthesis was performed using the Trinucleotide technology (ELLA Biotech) in order to exclude stop codons, methionine, cysteine, and asparagine.

Lengths between 8, 10, 12, 14, 16, and 20 amino acids were allowed, in which the last two amino acids were kept constant with the sequence Asp-Tyr for length 8–16 and Asp-Val for length 20. The design of the final two HCDR3 amino acids reflects human VDJ recombination. Short HCDR3s more often use J-fragment IGHJ4 with "DY" at the end of HCDR3, while longer HCDR3s (here 20 aa) more often use IGHJ6 with "DV" at the end of HCDR3.

Library inserts were generated by PCR using Phusion High-Fidelity DNA polymerase (NEB Biolabs). The resulting HCDR3 library inserts were ligated into the base templates, transformed into *E. coli* TG1F+ DUO (Lucigen) with a minimal library size of 3E+08 transformants per HCDR3 length, and phages were produced using VCSM13 helper phage (Agilent Technologies) using standard protocols.

## Phage display for isolation of PrP binders

Depending on the HCDR3 length, the library was divided into two sub-pools for panning: short (8–10 aa) and long (12–20 aa) HCDR3—that were run in parallel for the selection. First, two rounds of selection were performed by coating 96-well Maxisorp plates (Nunc) with decreasing amount of mouse $recPrP_{23-231}$ (1 and 0.5 μM, respectively, in PBS), overnight at 4°C. PrP-coated plates were washed and blocked with Superblock for 2 h. Input of $4 \times 10^{11}$ phages in 300 μl of PBS was used for the first round of panning. After 2 h blocking with Chemiblocker (Millipore), the phages were incubated with $recPrP_{23-231}$-coated wells for 2 h at room temperature (RT). The non-binding phages were then removed by extensive washing with PBS 0.05% Tween-20 (PBS-T) while mouse $recPrP_{23-231}$ bound phages were eluted with of 0.1 M Glycine/HCl, pH 2.0 for 10 min at RT, and the pH was neutralized by 1 M Tris pH 8.0. Eluted phages were used to infect exponentially growing Amber suppressor TG1F+ cells (Lubio Science). Infected bacteria were cultured in 2YT/Carbenicillin/1% glucose medium overnight at 37°C, 200 rpm and superinfected with VCSM13 helper phages. The production of phage particles was then induced by culturing the superinfected bacteria in 2YT/Carbenicillin/Kanamycin medium containing 0.25 mM isopropyl β-D-1-thiogalactopyranoside (IPTG), overnight at 22°C, 180 rpm. Supernatant containing phages from the overnight culture was used for the second panning round. Output phages from the second round were purified by PEG/NaCl precipitation, titrated, and selected in the following third rounds. For the third rounds either mouse $recPrP_{23-231}$, recPrP fragments or synthetic PrP peptides were used as bait at 0.25 μM in PBS for coating 96-well Maxisorp plates. Selection for binders to biotinylated PrP peptide was performed with the antigen in solution followed by capture on neutravidin-coated wells. Selection for $CC1_{23-50}$ binders was also performed in solid phase by capturing of $CC1_{23-50}$ biotinylated peptide on neutravidin-coated wells. After the third round of selection, DNA minipreps were prepared from the panning output pools by QIAprep Spin Miniprep kit (Qiagen) and the whole anti-PrP Fab enriched library was either used for NGS analysis or used for bacteria transformation.

## ELISA screening of bacterially expressed Fab phagemid vectors

Electrocompetent non-Amber suppressor MC1061 bacteria (Lubio Science) were transformed with pPD2-Fab phagemid vectors to perform the primary ELISA screening. DNA used for transformation derived from the following panning sets: full-length $recPrP_{23-231}$, $CC1_{23-50}$ in solid phase and in liquid phase, $N\text{-}OR_{39-66}$, $F\text{-}OR_{51-91}$ and $CC2\text{-}HC_{92-120}$. Single colonies were picked randomly by an automated colony picker and cultured in 384-well plate (Nunc) in 2YT/Ampicillin/1% glucose medium overnight at 37°C, 80% humidity, and 500 rpm. These precultures were used to prepare glycerol stock master plates. Expression plates were prepared by inoculating 2YT/Carbenicillin/0.1% glucose medium, followed by induction with 1 mM IPTG. After overnight culture at 25°C, 80% humidity, bacteria cultures were lysed for 1.5 h at 400 rpm, 22°C in borate buffered saline pH 8.2 containing EDTA-free protease inhibitor cocktail, 2.5 mg/ml lysozyme, and 40 U/ml benzonase. Fab-containing bacteria lysate was blocked with Superblock for 2 h and used for the ELISA screening. The following antigens were coated

on separate 384-well ELISA plates: anti-Fd antibody (The Binding Site GmbH) 1:1,000 in PBS, to check the expression level of each Fab clone in bacteria lysates, and mouse recPrP$_{23-231}$ at 87 nM in PBS. Antigen-coated ELISA plates were washed twice with TBS-T and blocked with Superblock for 2 h. Fab-containing bacteria lysates from the expression plate were transferred to corresponding wells of the ELISA plates. Wells containing only medium or POM antibodies at 50 nM in TBS-T were included as background and positive controls, respectively. After 2 h, ELISA plates were washed three times with TBS-T and anti-human F(ab')$_2$-alkaline phosphatase conjugated antibody (1:5,000 in TBS-T) was added. After 1 h incubation at RT, followed by three washings with TBS-T, Attophos substrate (Roche) was added and, after 10 min incubation, the ELISA signal (fluorescence) was measured. Fabs from bacteria lysates producing an ELISA signal 5–10 times higher than the technical background, which was calculated as the average of the coated well containing only medium, were considered as mouse recPrP$_{23-231}$ binder candidates. All the identified hits were checked in confirmatory ELISA screening. Anti- CC1$_{23-50}$ anti-OR$_{51-91}$, anti CC2-HC$_{92-120}$ and anti-GD binder hits were identified. Corresponding Fabs were subcloned into the Fab expression vector pPE2. pPE2-Fab plasmids were transformed in TG1F-chemicompetent cells and grown on LBagar/Kanamycin/1% glucose plates. Bacteria cultures of the selected clones were used for DNA minipreps followed by Sanger sequencing using the following sequencing primers: VH (5'-GATAAGCATGCGTAGGAGAAA-3') and M13Rev (5'-CAG GAA ACA GCT ATG AC-3').

### Affinity maturation of selected Fab3 and Fab71

Fab3 and Fab71 were further engineered to improve their affinity by using affinity maturation cassette libraries (Novartis) with either diversification in the HCDR2 or in the LCDR3. The HCDR2 and the LCDR3 sequence repertoires were diversified according to naturally occurring repertoire of rearranged human CDR sequences.

For LCDR3 libraries, primers were designed to incorporate up to 11 amino acids at defined ratios mimicking their natural occurrences: aspartic acid, glutamic acid, arginine, histidine, threonine, serine, glycine, alanine, leucine, valine, and tyrosine. Glutamine, proline and tryptophan were also allowed at certain positions of LCDR3. LCDR3 lengths of 9 and 10 amino acids were allowed for IGKV1–39 and IGKV3–15, in which the last threonine was kept constant.

For HCDR2 libraries, primers were designed to incorporate up to 10 amino acids at defined ratios mimicking their natural occurrences: aspartic acid, glutamic acid, arginine, histidine, threonine, serine, glycine, alanine, valine, and tyrosine. Isoleucine and tryptophan were also allowed at certain positions of HCDR3. A length of 17 amino acids was allowed for the HCDR2 of IGHV3–23 and IGHV3–30, in which the last seven amino acids were kept constant (YADSVKG).

For LCDR3 and HCDR2 maturation cassettes, the synthesis of randomized primer was performed using the Trinucleotide technology (ELLA Biotech) in order to exclude stop codons, methionine, cysteine and asparagine. The randomization of LCDR3 and HCDR2 was performed by two PCR steps using Phusion High-Fidelity DNA polymerase (NEB Biolabs). A first randomization PCR was performed using randomized primers (a forward primer to randomize LCDR3 and a reverse primer to randomize HCDR2). In a second

step the randomized fragments are amplified by using primers which also introduce restriction sites at both ends. This amplification PCR was then ligated into the base templates phagemid vectors using the restriction sites, transformed into *E. coli* TG1F+ DUO (Lucigen) with a minimal library size of 1E+08 transformants. For the HCDR2 affinity maturation libraries of Fab3 and Fab71, MfeI-HF and BssHII restriction digestion was performed to replace the parental HCDR2 sequence with randomized cassettes of diversified HCDR2 sequences. For the LCDR3 affinity maturation libraries of Fab3 and Fab71, MfeI and EcoRI restriction digested parental HCDR3 was grafted into the affinity maturation phagemid library with randomized LCDR3. HCDR2 and LCDR3 phagemid libraries of Fab71 and Fab3 were transformed into TG1F+ competent cells (Invitrogen), phage particles were produced by superinfection with VCSM13 helper phage and then precipitated by PEG/NaCl. These four phage libraries (Fab3 and Fab71 HCDR2; Fab3 and Fab71 LCDR3) were subjected to three rounds of panning by alternating exposure to mouse recPrP$_{23-231}$ and to the respective biotinylated peptides containing the targeted epitope (CC1$_{23-50}$ -biotinylated peptide for Fab3 and OR- biotinylated for Fab71). To drive the selection for high affinity binders, two strategies were used. As a standard condition for maturation, washing stringency was increased as compared to the panning conditions used for selection of the parental Fabs and the amount of antigen from one round to the next was decreased (from 200 nM at the first round to 1 nM at the third round). As an alternative approach, panning was performed in competition with 5-fold molar excess of the parental Fabs. After the third round of panning, DNA minipreps were used for NGS screening. Moreover, reading frames of the enriched pools of matured Fab3 and Fab71 were inserted into pPE2 vector for expression in bacteria, followed by ELISA screening. For each Fab, four output pools were obtained. For each pool, 368 colonies were randomly picked and transferred to a 384-well plate for primary ELISA screening using bacteria lysates containing soluble Fabs, as described above. To differentiate matured Fabs with higher affinity as compared to the parental version, off-rate and on-rate ELISA screens were performed using mouse recPrP$_{23-231}$ coated ELISA 384-well plates. Briefly, for off-rate selection, Fab-containing *E. coli* lysates were exposed for 2 h to recPrP$_{23-231}$ coated wells and, after ten washing steps lasting from 1 h to overnight, anti-human F(ab')$_2$-alkaline phosphatase conjugated antibody was added (1:5,000 in TBS-T), followed by Attophos substrate (Roche) and measurement of the ELISA signal (fluorescence) as described above. During the on-rate ELISA screen, the incubation time of the diluted Fab-containing *E. coli* lysates with recPrP$_{23-231}$ coated wells was only 20 min, followed by three fast washes. From the primary screening 95 positive clones were sequenced and assessed in confirmatory screens by off-rate and on-rate ELISA whereby retention of epitope specificity was checked by measuring the binding of affinity-matured versions of Fab3 and Fab71 to CC1$_{23-50}$ and F-OR$_{51-90}$ biotinylated peptides. Hits consisting of 19 Fab3 matured clones and 5 Fab71 matured clones were selected and purified by IMAC.

### Sample preparation for NGS

Polyclonal DNA minipreps isolated from the third panning output pools were used as PCR template to amplify the HCDR3 of the

selected Fabs and add the adapters required for sequencing on Illumina sequencer MiSeq. The PCR protocol has been described (Liu *et al*, 2019).

The PCR product was purified and DNA concentration was measured using the Qubit DNA High sensitivity kit (Invitrogen). Samples were analyzed on a MiSeq using MiSeq reagent kit *MiSeq v2 Reagent kit 300 cycles PE.*

## NGS data analysis

The data analysis of the NGS FastQ output files was performed as described (Liu *et al*, 2019). For each panning output, 100,000 sequences were analyzed using the fixed flanking sequences on the boundary of variable region as template to locate and segment out the HCDR3 sequence. ~40,000 to 70,000 HCDR3 sequences were identified depending on the panning output pools, and included into frequency reports in CSV format.

For determination of clones to high immunogenic PrP epitopes, we selected HCDR3 displaying ≥ 20 NGS counts in recPrP$_{23-231}$ panning in 100,000 analyzed sequences. For rare clones against less immunogenic PrP epitopes, HCDR3 were identified according to the following criteria: NGS count in recPrP$_{23-231}$ panning = 1 and, to avoid selecting for sequences resulting from PCR or sequencing errors, sum of the NGS counts across all the panning outputs ≥ 10.

## Rescue of clones identified in NGS

Fab clones of interest based on the NGS-binding profile can be retrieved from the polyclonal DNA output after phage panning by an assembly PCR approach. For FabRTV, first two separate PCR reactions were executed to amplify the Fab in two fragments, one ranging from VL until HCDR3 (LC –HCDR3 PCR) and the second one ranging from HCDR3 until the tag (HCDR3-CH1-tag). Primers specific for FabRTV HCDR3 sequence were used (FabRTV_HCDR3_fw: 5'-TCGTTACGTTCGTGGTTACGGTTCTCC-3'; FabRTV_HCDR3_rw: 5'-GGAGAACCGTAACCACGAACGTAACGA-3' in combination with PAL_fw (PAL_fw: 5'-GGAAACAGCTATGACCATGATTACGCCAAG-3' and Flag_rw (Flag_rw: 5'- CGCACCTTTGTCATCGTCATCTTTATAGTCG- 3'). The polyclonal DNA pool from the third round was taken as template. LC–HCDR3 and HCDR3-CH1-tag amplicons were generated using Phusion High-Fidelity DNA Polymerase and the following conditions: 98°C for 30 s, 98°C for 10 s, 62–65°C for 30 s, 30 cycles at 72°C for 100 s (LC –HCDR3 PCR) or 72°C for 40 s (HCDR3-CH1-tag), and 72°C for 5 min. PCR products were purified. LC –HCDR3 and HCDR3-CH1-Flag fragments were then assembled in the next PCR using the primers PAL_fw and Flag_rw. The assembly PCR was performed using 60 ng LC-HCDR3 and 20 ng HCDR3-CH1-Flag as template in two steps. A first PCR sequence was run without primers (98°C for 30 s, 98°C for 10 s, 55°C for 30 s, five cycles at 72°C for 100 s, and 72°C for 7 min). Then, primers PAL-fw and Flag-rw were added and the second PCR step was run (98°C for 30 s, 98°C for 10 s, 60°C for 30 s, 25 cycles at 72°C for 120 s, and 72°C for 7 min). The assembly PCR product was purified, digested by NruI and EcoRI and ligated into pPE2 expression plasmid. Retrieved clones were transformed into XL1-Blue electrocompetent cells and DNA was prepared from isolated colonies for Sanger sequencing.

## Expression and purification of selected anti-PrP Fabs

Chemical competent BL21(DE3) cells (Invitrogen) were transformed with selected pPE2-Fab plasmids and grown on LBagar/Kanamycin/ 1% glucose plates. A single colony was inoculated into 20 ml of 2xYT/Kanamycin/1% glucose pre-culture medium and incubated for at least 4 h at 37°C, 220 rpm. One liter of 2YT-medium containing Kanamycin/0.1% glucose was inoculated with 20 ml pre-culture and Fab expression was induced by 0.75 mM IPTG followed by incubation overnight at 25°C, 180 rpm. The overnight culture was centrifuged at 4,000 *g* at 4°C for 30 min and the pellet was frozen at −20°C. For Fab purification, thawed pellet was resuspended into 20 ml lysis buffer: 0.025 M Tris pH 8; 0.5 M NaCl; 2 mM MgCl$_2$; 100 U/ml Benzonase (Merck); 0.25 mg/ml lysozyme (Roche), EDTA-free protease inhibitor (Roche), and incubated for 1 h at RT at 50 rpm. Lysate was centrifuged at 16,000 *g* at 4°C for 30 min and supernatant was filtrated through 0.22 μm Millipore Express® Plus Membrane. Fab purification was achieved via the His6-Tag of the heavy chain by IMAC. Briefly, after equilibration of Ni-NTA column with running buffer (20 mM Na-phosphate buffer, 500 mM NaCl, 10 mM Imidazole, pH 7.4), the bacteria lysate was loaded and washed with washing buffer (20 mM Na-phosphate buffer, 500 mM NaCl, 20 mM Imidazole, pH 7.4). The Fab was eluted with elution buffer (20 mM Na-phosphate buffer, 500 mM NaCl, 250 mM Imidazole, pH 7.4). Buffer exchange in PBS was performed using PD-10 columns, Sephadex G-25M (Sigma).

## ELISA for epitope specificity of purified Fabs

For epitope profiling of the Fabs by ELISA, 384-well SpectraPlates (Perkin Elmer) were coated with the following antigens at 87 nM in PBS, at 4°C overnight: mouse recPrP$_{23-231}$, recPrP$_{23-110}$, recPrP$_{90-231}$, recPrP$_{121-231}$, BSA and neutravidin. Plates were washed three times in PBS-T and blocked with 50 μl per well of Superblock for 2 h at RT. Biotinylated PrP peptides CC1$_{23-50}$, N-OR$_{39-66}$, F-OR$_{51-91}$ and CC2-HC$_{92-120}$ were captured on neutravidin-coated wells for 1 h at RT. Purified Fabs at 150 nM in PBS-T were then incubated for 2 h at RT. After three washing with PBS-T, binding Fabs were detected by anti-human F (ab')$_2$-alkaline phosphatase conjugated antibody (1:1,000 in PBS-T). After 1 h incubation at RT, followed by three washings with PBS-T, Attophos substrate (Roche) was added and, after 10 min incubation, the ELISA signal was measured. The relative affinities of Fab binding to PrP were determined by using the Fabs at serial dilutions by measuring the concentration of Fab required to achieve 50% maximal binding (EC50). Data analysis was performed using Non-linear regression (GraphPad Prism, GraphPad Software).

## Competition ELISA for epitope mapping

The approach described by Polymenidou *et al* (2008) was used. A library of overlapping 12-mer mouse PrP peptides, shifted by two amino acids, was synthesized by Jerini, Germany. It consisted of 50 peptides, spanning the whole FT of the mouse PrP sequence. Mouse recPrP$_{23-231}$ at 65 nM in PBS was coated to 384-well HB plates (Perkin Elmer), overnight at 4°C. Plates were washed three times with PBS-T and blocked with Superblock for 2 h at RT. After washing, plates were incubated with 400 nM of anti-PrP Fabs in 1% Superblock in PBST, with or without 175 molar excess of each

dodecameric peptide (final concentration of 70 μM). After 2 h at RT plates were washed three times and incubated with anti-human Fab specific-HRP conjugated secondary antibody (Sigma, 1:1,000 in PBS-T) for 1 h at RT. TMB, was added, incubated for three minutes at RT, and the chromogenic reaction stopped by addition of 0.5 M $H_2SO_4$. The absorbance at 450 nm was measured in a plate reader (Perkin Elmer, EnVision).

### ELISA with dilution series for EC50 determination

To compare affinity-matured and parental Fabs, ELISA 384-well HB plates (Perkin Elmer), were coated with mouse recPrP$_{23-231}$ at 87 nM in PBS overnight at 4°C. After three washing steps in PBS-T and blocking with Superblock for 2 h at RT, plates were incubated for 2 h at RT with 1:2 serially diluted Fabs at starting concentration of 150 nM in PBS-T. Plates were incubated with anti-human Fab specific-HRP conjugated secondary antibody (Sigma, 1:1,000 in PBS-T) for 1 h at RT, followed by washing and development of chromogenic reaction by TMB, as described above. The absorbance at 450 nm was measured in a plate reader (Perkin Elmer, EnVision).

For Fabs that were identified by mining of NGS datasets of human antibody repertoires, human recPrP$_{23-230}$ at 87 nM in PBS was used for coating and serially diluted Fabs were tested starting from 3 μM in PBS-T.

### Measurement of binding kinetics by surface plasmon resonance (SPR)

Binding kinetic of the purified Fabs to full-length PrP was monitored at 25°C using ProteOn XPR36 surface plasmon resonance biosensor (Bio-Rad Laboratories). Mouse recPrP$_{23–231}$ was immobilized on a carboxymethylated-dextran sensor chip (CM5, Bio-Rad) by EDC/NHS chemistry at either low or high density (1,000 and 100 resonance units (RU), respectively) in the test flow cells. Two flow cells were left untreated, one was exposed to activation and deactivation buffers used in EDC/NHS chemistry and one was used to immobilize BSA as a control. Six serial 1:2 dilution of Fab at starting concentration of 100 nM were prepared in SPR running buffer (10 mM Tris/HCl buffer pH 8.5, 500 mM NaCl, 3 mM EDTA, 0.05% Tween-20) and injected at flow rate of 100 μl/min for 120 s. Association and dissociation were monitored over a 300 s interval. BSA immobilized flow cell was used as negative control for the antigen and responses were corrected by referring to the flow cell where only SPR running buffer was injected. The sensor chip was regenerated after each round of Fab injection by flowing the sensor chip chambers with 20 nM NaOH for 60 s at 25 μl/min. Sensorgrams were fitted by Langmuir equation (the 1:1 interaction model) and antibody binding constants ($k_{on}$ and $k_{off}$ and $K_D$) were calculated by the Bio-Rad ProteOn manager software.

### Epitope confirmation

Surface plasmon resonance approach was used to confirm the epitope specificity of the different anti-PrP Fabs. An NLC sensor chip (Bio-Rad) was used to immobilize the biotinylated PrP peptides CC1$_{23-50}$, N-OR$_{39-66}$, FL-OR$_{51-91}$ and CC2-HC$_{92-120}$ in different flow cells at 700 RU. Dilution series of each Fab were injected at 100 μl/

min starting from 100 nM and real time binding was monitored (120 s association time and 300 s dissociation time, regeneration by 20 nM NaOH for 60 s at 25 μl/min).

### Establishment of CAD5 Prnp$^{−/−}$ cells by CRISPR

The CAD5 cell line derived from Cath.a-differentiated (CAD) cells was reported to be responsive to several prion strains (Mahal *et al*, 2007). For control in our experiments, CAD5 *Prnp*$^{−/−}$ cells were established by CRISPR/Cas9-mediated gene ablation. Briefly, a guide RNA (gRNA) targeting exon 3 of the *Prnp* gene (5′- TCA GTC ATC ATG GCG AAC CT -3′) was cloned into the pKLV-U6gRNA(BbsI)-PGKpuro2ABFP vector (Addgene 50946,) to generate pKLV-*Prnp* sgRNA plasmid. A CAD5 cell line stably expressing Cas9 (CAD5-Cas9) was generated by transfecting linearized pCMV-hCas9 (Addgene 41815). CAD5-Cas9 cells were then transfected with pKLV-*Prnp* sgRNA by Lipofectamine 2000 (Invitrogen, 11668-019). Transfected cells were selected and enriched by puromycin (2 μg/ml) and single cell clones were obtained by limited dilution. Cell clones were grown and expanded. To confirm gene editing of the *Prnp* gene in cell clones, a region embracing gRNA was PCR-amplified using a forward primer (5′-TGC AGG TGA CTT TCT GCA TTC TGG-3′) and reverse primer (5′-GCT GGG CTT GTT CCA CTG ATT ATG GGT AC-3′), the PCR product was cloned into pCR-Blunt II-TOPO vector (Invitrogen, 45-0245) and sequenced to verify the frameshifted indels. To confirm gene editing of the *Prnp* gene in cell clones, a region embracing gRNA was PCR-amplified using a forward primer (5′-TGC AGG TGA CTT TCT GCA TTC TGG-3′) and reverse primer (5′-GCT GGG CTT GTT CCA CTG ATT ATG GGT AC-3′), the PCR product was cloned into pCR-Blunt II-TOPO vector (Invitrogen, 45-0245) and sequenced to verify the frameshifted indels.

### Flow cytometry

Binding of Fabs to wtPrP$^C$ on the surface of CAD5 PrP$^{+/+}$ cells was determined by flow cytometry. CRISPR/Cas9-derived CAD5 *Prnp*$^{−/−}$ cells were used as negative control. Briefly, Confluent CAD5 *Prnp*$^{+/+}$ and CAD5 *Prnp*$^{−/−}$ cells in 75-cm flasks were washed with PBS, detached in dissociation buffer (2 mM EDTA, PBS) and washed twice in ice-cold FACS buffer (10 mM EDTA, 2% FBS in PBS) by centrifugation at 190 *g* for 5 min at 4°C. Cells were stained with Trypan Blue, counted and density of live cells adjusted to 0.5 × 10$^6$ cell in 100 μl. Hundred microliter of cell suspension were added to each well of a 96-well round bottom tissue culture plate, and centrifuged at 800 *g* for 2 min at 4°C. Cells were incubated for 30 min at 4°C with the Fabs at 150 μg/ml. After washing twice in FACS buffer, Fabs bound to cell-surface wtPrP$^C$ was detected by incubating the cells with APC-labeled anti-human Fab secondary antibody (Jackson) diluted 1:400 for 30 min at 4°C in the dark. Cells stained with anti-PrP POM1 Fab and APC-labeled anti-mouse Fab were used as a positive control while staining with secondary antibodies was used to establish the background fluorescence. Cells were washed three times with ice-cold FACS buffer and transferred into micro-FACS tubes. Data were acquired using FACS-Canto II cytometer (BD Biosciences, San Jose, CA, USA) and the geometric mean fluorescence (MFI) of stained cells was measured using FlowJo software.

## Immunohistochemistry

Staining was performed on sections from brain tissues fixed in formalin and embedded in paraffin. After deparaffinization through graded alcohols and heat-induced antigen retrieval in citrate buffer (0.01 M; pH 6), sections were blocked and incubated with the anti-PrP Fabs at 12 µg/ml and antibody for Microtubule Associated Protein 2 (MAP2; 1:500, Abcam). Double staining was performed with fluorescently labeled secondary antibodies (goat anti-Human IgG F(ab')2:Tritc 1:150, Bio-Rad and goat anti-rabbit 1:500, Alexa Fluor 488, Invitrogen), followed by nuclear staining by DAPI (Life technologies). Images acquisition was done by using the fluorescence microscope (BX-61; Olympus), equipped with a cooled black/white charge-coupled device camera, using identical acquisition settings. Images were analyzed using the image-processing software CellF.

## Immunoprecipitation

For immunoprecipitation of PrP from brain extracts, the tissue was homogenized in ice-cold IP buffer (75 mM NaCl, 1% Igepal, protease inhibitor mixture (Sigma), 50 mM Tris-Cl, pH 7.4). After centrifugation at 1,000 $g$ for 5 min, the supernatant was recovered and protein content quantified by BCA assay (Pierce). One milliliter of Dynal sheep-anti-mouse IgG paramagnetic beads were non-covalently coupled with 60 µg of anti-His mAb (Invitrogen) in PBS plus 0.1% immunoglobulin-free BSA (Sigma) for 1 h at RT on a rotating wheel. Three molar excess of the His-tagged Fab were added to allow the formation of the complex with the anti-His antibody-coupled dynabeads and after 1 h incubation, three washes in PBS plus 0.1% immunoglobulin-free BSA were performed. Five hundred microgram of brain homogenate was diluted in IP buffer and incubated first with 20 µl of dynabeads coupled only to the anti-His antibody without the Fab to remove the unspecific binding of PrP to the beads and to the IgG. Then the cleared brain homogenate was incubated with 50 µl of Fab-anti-His antibody-coupled dynabeads and immunoprecipitation was performed for 2 h at RT on a rotating wheel. After five washes with 150 mM NaCl, 0.5% Igepal, 50 mM Tris-Cl, pH 7.4, elution of immunoprecipitated PrP was performed by incubation for 3 h at 4°C with 62 µg of 12-mer PrP peptides in 40 µl of PBS plus protease inhibitors. Peptide P1: KKRPKPGGWNTG was used for competition with Fab3 and peptide P17 TWGQPHGGGWGQ was used for competition with Fab71 and Fab100. Peptides P2 RPKPGGWNTGGS and P15 H-PQGGTWGQPHGG-OH were used as negative controls. Eluate was finally collected and supplemented with loading buffer (NuPAGE, Invitrogen) for western blot analysis using the mouse monoclonal antibody POM1 and the rabbit polyclonal antibody XN (1:5,000, produced in-house) for PrP detection.

## Immunoblot analysis

For epitope confirmation of the Fabs by western blot, brain from transgenic mice expressing different PrP deletion mutants was homogenized in 10 volumes of lysis buffer (50 mM Tris–HCl pH 8, 0.5% Na deoxycholate, and 0.5% Igepal, protease inhibitors (complete Mini, Roche) using TissueLyser LT for 5 min for two cycles. After centrifugation at 1,000 $g$ for 5 min at 4°C to remove debris, protein concentration in the post-nuclear supernatant was measured by BCA. Twenty five microgram of total proteins were separated by SDS–PAGE (Novex

NuPAGE 12% Bis-Tris Gels) and transferred to PVDF membrane. Membranes were blocked with 5% milk in TBS-T for 1 h at RT and incubated overnight at 4°C with anti-PrP Fabs at 10 µg/ml. After washing, the blots were incubated with secondary antibody horseradish peroxidase (HRP)-conjugated goat anti-human Fab IgG (H+L; Sigma, A0293) diluted 1:10,000 in blocking buffer for 1 h at RT. Blots were developed using Luminata Crescendo Western HRP substrate (Millipore) and visualized using the Stella system (model 3200, Raytest).

For PrP<sup>sc</sup> detection in prion-infected cultured organotypic cerebellar slices (COCS), cerebellar slices were washed in PBS and then scraped off the membrane using 1 ml of ice-cold PBS, pelleted and homogenized by trituration in lysis buffer. Samples were centrifuged at 1,000 $g$ for 5 min at 4°C and protein concentration in the post-nuclear supernatant determined by BCA.

Samples were adjusted to 20 µg protein in 20 µl and digested with 5 µg/ml proteinase-K (PK) in digestion buffer (0.5% wt/vol sodium deoxycholate and 0.5% vol/vol Nonidet P-40 in PBS) for 30 min at 37°C. PK digestion was stopped by adding loading buffer (NuPAGE, Invitrogen) and boiling samples at 95°C for 5 min. Proteins were separated on a 12% Bis-Tris polyacrylamide gel and blotted onto a PVDF membrane by using the iblot apparatus (Bio-Rad). Membranes were blocked with 5% top-block in PBS-T followed by incubation with POM1 mouse IgG$_1$ antibody (200 ng/ml). After washing, secondary antibody rabbit anti–mouse IgG$_1$ (1:10,000, Zymed) used was. For actin detection, mouse monoclonal anti-Actin (1:10,000, MAB1501R Merck, Millipore) was used. Blots were developed Luminata Crescendo Western HRP substrate (Millipore) and visualized using the Fuji Stella (Bio-Rad).

## Antibody treatment in cultured organotypic cerebellar slices

Cultured organotypic cerebellar slices were prepared from 9–12 days old tga20 pups as previously described (Falsig et al, 2012). For prion experiments, COCS were infected as free-floating sections with 100 µg per 10 slices of RML6 (Rocky Mountain Laboratory strain mouse-adapted scrapie prions at six passage) brain homogenate from terminally sick prion-infected mice. As control, non-infectious brain homogenate (NBH) from CD1-inoculated mice was used. After incubation with brain homogenates diluted in physiological Grey's balanced salt solution for 1 h at 4°C, the slices were washed and 5–8 sections were seeded on a 6-well PTFE membrane insert. Treatment with Fab (550 nM) was started 1 day after plating and supplied at every medium exchange. At 45 days in culture, slices were fixed and processed for immunocytochemistry.

## Immunofluorescence staining of COCS and NeuN morphometry

For immunofluorescence staining, COCS were washed twice in PBS and fixed in 4% formalin overnight at 4°C. After washing in PBS, the COCS were incubated with blocking buffer (0.05% Triton X-100 vol/vol, 0.3% goat serum vol/vol in PBS) for 1 h at RT and then incubated for 3 days at 4°C with the monoclonal mouse anti-NeuN antibody conjugated with Alexa-488 (1:1,000, clone A60, Life Technologies) at a concentration of 0.5 µg/ml into blocking buffer. Slices were then washed two times with PBS for 15 min and then incubated with 4′,6-diamidino-2-phenylindole (DAPI; 1 µg/ml) in PBS at RT for 30 min to visualize cell nuclei. Two subsequent washes in PBS were performed and COCS were mounted with fluorescence

mounting medium (DAKO) on glass slides. NeuN morphometry was performed by image acquisition on a fluorescence microscope (BX-61, Olympus) at identical exposure times. Morphometric analysis was performed to quantify the area of immunoreactivity on unprocessed images using a custom written script for cell^P (Olympus).

### Prion-infected CAD5 cells

CAD5 $Prnp^{+/+}$ and CAD5 $Prnp^{-/-}$ cells were cultured in phenol red free OPTI-MEM supplemented with 10% FBS, Glutamax, penicillin G and streptomycin at 37°C in 5% $CO_2$/95% air. CAD5 $Prnp^{+/+}$ and CAD5 $Prnp^{-/-}$ cells ($5 \times 10^4$ in 2 ml of medium) were seeded into 6-well plates (Corning Costar) and cultured for 1–2 days before exposure to 500 µg/ml of prion-infected mouse brain homogenate. Non-infectious brain homogenate (NBH) from CD1 mice at the same dilution was used as control. Treatment with Fabs at 10 µg/ml (200 nM) was initiated 2 h after infection and repeated at every split by spiking into the culture medium. Three biological replicates were prepared for each condition. The inoculum was removed after 3 days and the cells were split 1:5 every 3–4 days. After four splits (14 days *in vitro*, DIV) the cells were assayed for $PrP^{Sc}$ by the TR-FRET assay as described below.

### Quantification of $PrP^{Sc}$ by homogenous-phase TR-FRET

Prion-infected CAD5 $Prnp^{+/+}$ and CAD5 $Prnp^{-/-}$ cells, and NBH-inoculated control cells, at 14 DIV in 6-well plates, were detached by pipetting and $10^4$ cells in 40 µl of medium were seeded per well of a 384-well plated (Perkin Elmer) and cultured for one additional day. To detect $PrP^{Sc}$ selectively in homogenous phase, all the following steps were performed by sequentially adding the reagents into each well without any washing step. First, proteinase-K (PK) digestion was carried out by adding 10 µl of lysis buffer containing 50 or 25 µg/ml of PK (10 or 5 µg/ml PK final concentration) and plates were incubated at 37°C for 1.5 h at 100 *g*. PK digestion was stopped by adding 4 µl/well of PMSF (final concentration 2 mM) and incubated for 10 min at RT. A denaturation step was performed by NaOH at 58 mM for 10 min at RT to enhance the accessibility of buried epitopes in $PrP^{Sc}$. After neutralization by $NaH_2PO_4$ buffer (58 mM, pH 4.3) for 10 min at RT, plates were processed for FRET measurement. To detect PK-resistant $PrP^{Sc}$, anti-PrP holo-antibodies POM19 and POM1 recognizing two close epitopes in the globular domain (Polymenidou *et al*, 2008) were used. As described (Ballmer *et al*, 2017), first, 5 µl/well (2.5 nM final) POM19 (made in-house), coupled to Europium (Eu), and diluted in 1X Lance Detection Buffer (Perkin Elmer) was dispensed. Then, 5 µl (5 nM final) POM1 conjugated to Allophycocyanin (APC) was added. TR-FRET readout was performed after 1 h incubation at 4°C using an EnVision 2105 Multimode Plate Reader (PerkinElmer) with previously defined measurement parameters (Ballmer *et al*, 2017). The collected fluorescence data were corrected by both background and spectral overlap between Eu and APC channel. Net FRET calculations and blank subtractions were performed as previously described (Ballmer *et al*, 2017).

### Identification and cloning of Fabs from human antibody repertoire datasets

NGS datasets of antibody repertoires from published human healthy donors, here referred to as DW, BB and DK (DeKosky *et al*, 2016;

DeWitt *et al*, 2016; Briney *et al*, 2019). Sequencing data included naïve and memory B cells of 14 donors in total that could be downloaded already pre-processed. HCDR3 amino acid sequences harboring the SYGGY sequence of Fab71 were analyzed and selected if the overall difference to Fab71 HCDR3 was ≤ 3 residues. Afterward, HCDR3 duplicates, with the same VH, DH and JH segments, found in technical replicates were removed. Then, unique Fab71 similar HCDR3 in each donor were listed and compared across all the subjects from all the datasets. This identified HCDR3 similar to Fab71 that were present in more than a donor with different VH, DH, and JH segments. Nucleotide sequences of the variable regions, including each selected HCDR3 Fab71 similar along with the VH, DH, and JH segments reported in the sequencing datasets, were synthesized (Genescript) and cloned into pPE2 expression vector by restriction digestion.

### High-throughput antibody profiling in unselected patient cohort

Small volumes (< 100 µl) of plasma samples were obtained from the Institute of Clinical Chemistry at the University Hospital of Zurich as unique biospecimens. In order to test the samples for the presence of IgGs reactive against human $recPrP_{23-230}$, high-binding 1536-well plates (Perkin Elmer, SpectraPlate 1536 HB) were coated with 1 µg/ml human $recPrP_{23-230}$ in PBS at 37°C for 1 h, followed by three washes with PBS-T and by blocking with 5% milk in PBS-T for 1.5 h. Three microliter plasma, diluted in 57 µl sample buffer (1% milk in PBS-T), were dispensed at various volumes into human $recPrP_{23-230}$ coated 1,536-well plates using contactless dispensing with an ECHO 555 Acoustic Dispenser (Labcyte). Thereby, dilution curves ranging from plasma dilutions 1:50 to 1:6,400 were generated (eight dilution points per patient plasma sample). After the sample incubation for 2 h at RT, the wells were washed five times with wash buffer and the presence of IgGs bound to the human $recPrP_{23-230}$ was detected using an HRP-linked anti-human IgG antibody (Peroxidase AffiniPure Goat Anti-Human IgG, Fcγ Fragment Specific, Jackson, 109-035-098, at 1:4,000 dilution in sample buffer). The incubation of the secondary antibody for one hour at RT was followed by three washes with PBS-T, the addition of TMB, an incubation of three minutes at RT, and the addition of 0.5 M $H_2SO_4$. The well volume for each step reached a final of 3 µl. The plates were centrifuged after all dispensing steps, except for the addition of TMB. The absorbance at 450 nm was measured in a plate reader (Perkin Elmer, EnVision) and the inflection points of the sigmoidal binding curves were determined using a custom designed fitting algorithm. Samples reaching half-maximum saturation (shown as the inflection point of the logistic regression curve) at a concentration ≤ 1:100, i.e. at −log(EC50) ≥ 2, and with a mean squared residual error < 20% of the actual −log(EC50) were considered hits. The inclusion of a threshold for fitting error ensures a reliable identification of positives from high-throughput screening. For the validation screen, hits from HTS were tested against a panel of antigens consisting of human $recPrP_{23-230}$, bovine serum albumin (BSA, Thermo Scientific), and human Apolipoprotein E ε3 (Peprotech), at 1 µg/ml for all antigens. An additional uncoated condition was included to account for residual binding to either the blocking buffer or to the plate. The validation screen was performed identically to the HTS, except that triplicates were used instead of unicates. Samples from the validation screen were considered confirmed if

−log(EC50) for PrP ≥ 2 (distinct reactivity against PrP) and −log (EC50) for other targets < 2 (no distinct reactivity against any other control target). All data, including the patient-associated demographic and medical data, was stored in a MS-SQL database.

### Kappa and Lambda light-chain ELISA for total IgG and anti-PrP autoantibodies from plasma

To test total IgG from plasma, 384-well HB plates (Perkin Elmer) were coated with AffiniPure Goat anti-human IgG, Fc gamma fragment antibody, at 1 μg/ml in PBS for 2 h at 37°C. To test anti-PrP autoantibodies, plates were coated with human recPrP$_{23-230}$. After washing and blocking in 5% Milk in PBS-T for 90 min at RT, serial dilutions of patient plasma (starting from 1:8,000 and 1:50 dilution for total IgG and anti-PrP autoantibody ELISA, respectively) were incubated for 2 h at 37°C. Pooled normal human plasma (Innovative Research, #IPLA-N), IgG-Kappa, IgG-Lambda at a concentration of 200 ng/ml or humanized POM1, (hPOM1 at 200 ng/ml and 2 μg/ml for total IgG and anti-PrP autoantibody ELISA, respectively) served as controls. After washing five times, the following HRP-coupled secondary antibodies were incubated at 1:4,000 in PBS-T for 1 h, at RT: Goat anti-human Kappa-HRP, goat anti-human Lambda-HRP or goat anti-human IgG Fc-gamma specific-HRP (Jackson ImmunoResearch #109-035-098). The plates were developed as described above.

### IgG purification from human blood samples

For further validation, 50–250 μl of plasma were used for immunoglobulin purification via 1:1 mixture of proteinA/proteinG resin (GE Healthcare). Slurry was washed once with PBS. Blood samples were spun down at 16,000 g for 5 min, mixed with 10% of slurry and incubated at 4°C o/n on a rotating wheel. Slurry-blood mix was transferred to screw cap spin columns (Pierce) and beads were washed twice with PBS. Beads were spun down one more time to remove residual PBS and 100 μl of 0.1 M glycine, pH 1.8 was added to the beads. Four hundred microliter of 1 M Tris, pH 8.0 were added to the bottom of the spin columns for neutralization of elution buffer after centrifugation. For buffer exchange, 0.5 ml Centrifugal Filters (Amicon) were used. Protein concentration was measured using NanoDrop ND-1000 Spectrophotometer.

IgG fractions of blood samples were diluted at 100 μg/ml in 1% skim milk in PBS-T and assessed for binding to human recPrP$_{23-230}$, murine recPrP$_{23-110}$ and human recPrP$_{121-230}$ and BSA (coated at 43 nM) in ELISA using goat anti-human IgG Fc-gamma specific-HRP (Jackson ImmunoResearch #109-035-098), 1:4,000 for detection.

### Statistical analysis

For *in vitro* and *ex vivo* experiments (COCS), sample sizes were chosen according to the standard practice for the related methodologies. For the antibody screen from human plasma, sample size was limited by the patient's sample availability. In all the experiments using COCS and CAD5 cells, cells and slices were randomized to the exposure with either prion or NBH samples and the treatment with the different Fabs or untreated (only vehicle, PBS).

Continuous variables were described as means ± standard error of the mean (sem) for the number of replicates indicated in the legends of each figure. Categorical variables were expressed as counts.

### The paper explained

#### Problem
Antibody-based immunotherapy might be an efficient way for the treatment of prion diseases, for which no cure exists. However, antibodies targeting specific regions of the prion protein (PrP) can either be neurotoxic or neuroprotective. Hence, the identification of anti-PrP antibodies specifically targeting neuroprotective epitopes is of high importance for the generation of safe prion immunotherapeutics.

#### Results
In this study, we used Fab phage display to retrieve antibodies targeting different epitopes of PrP from a synthetic human antibody library. We demonstrated that antibodies targeting the N-terminal part of PrP were neuroprotective in a model of prion-induced neurodegeneration. Interestingly, mining of published human antibody databases confirmed the presence of such anti-PrP antibodies in naïve repertoires of circulating B cells from healthy humans. We also found high-titer PrP autoantibodies directed against the flexible tail of PrP in the plasma of unselected hospitalized patients without any clinical features of a pathological disease.

#### Impact
Our data demonstrate the presence of naturally occurring, innocuous anti-PrP antibodies in humans which may constitute a potential source for the development of effective and safe immunotherapeutics to combat prion diseases.

Python and R software as well as GraphPad Prism version 5 were used for data visualization and statistical testing. The statistical test used is indicated in each figure legend. In most of the cases, we used analysis of variance ANOVA to compare more than two groups of normally distributed data. The specific *post hoc* analysis that was used is indicated in each figure legend. Median test was used to assess the statistical difference of age. For categorical data (ICD codes and gender), we used the chi-square test of independence of variables in a contingency table with minimum five frequencies in each cell. For multiple comparisons, Bonferroni correction was applied.

## Data and software availability

Phage display NGS datasets and codes for the analysis of data from the high-throughput screenings are provided on github: https://github.com/AndraCh/HTS_analysis.

**Expanded View** for this article is available online.

### Acknowledgements
We thank Rita Moos and Cinzia Tiberi for support in recombinant antigens and Fab purification, Irina Abakumova for TR-FRET antibody-fluorophore conjugation, and Stefan Schauer for the SPR experiments. We thank Dr. Mario Nuvolone for helpful discussions. We also thank the hospital patients supporting research, especially the individuals who generously agreed to grant us a second blood donation. We acknowledge the help provided by the Clinical Trial Center (CTC), University Hospital of Zurich, and the Institute of Clinical Chemistry, University Hospital of Zurich. AA is the recipient of an Advanced Grant of the European Research Council (grant No 670958) and is supported by grants from the Swiss National Foundation (SNF; grant No 179040 and 183563), the Clinical Research

Priority Programs "Small RNAs" (https://www.uzh.ch/en/research/medicine/clinic.html), SystemsX.ch (grant No PrionX 2014/260 and SynucleiX 2015/320), the ERA-net (grant No 160672), the Swiss Personalized Health Network (grant No SPHN, 2017DRI17), the European Research Council (Proof of Concept, grant No 768415) and by a Distinguished Investigator Award of the Nomis Foundation and a donation from the estate of Dr. Hans Salvisberg. SH received support from a grant from SystemX.ch (grant No SynucleiX 2015/320). AS and SS are recipients of the Career Development Award grant from the Stiftung Synapsis - Alzheimer Forschung Schweiz AFS. KF received unrestricted support by Ono Pharmaceuticals and was funded by the Theodor Ida Herzog-Egli Stiftung. RR was supported by a Career Development Award from the Stavros Niarchos Foundation.

## Author contributions

Conceived and designed the experiments: AS, NG, and AA. Performed the experiments: AS, KF, ME, GH, JG, SF, ML, and RR. Analyzed the data: AS, NG, AC, ME, and SS. Contributed reagents/materials/analysis tools: NG, SE, TP, CZ, SH, and AA. Wrote the paper: AS, SH, and AA.

## Conflict of interest

The authors declare that they have no conflict of interest.

## For more information

- Authors webpage: http://www.neuropathologie.usz.ch and https://www.novartis.com/our-science/novartis-institutes-biomedical-research
- Information for persons with CJD: https://cjdfoundation.org
- Information for persons interested in prion research: https://www.cureffi.org

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
