## [Review Process File · EMBO Molecular Medicine]

Protective anti-prion antibodies in human immunoglobulin repertoires

Assunta Senatore, Karl Frontzek, Marc Emmenegger, Andra Chincisan, Marco Losa, Regina Reimann, Geraldine Horny, Jingjing Guo, Sylvie Fels, Silvia Sorce, Caihong Zhu, Nathalie George, Stefan Ewert, Thomas Pietzonka, Simone Hornemann, and Adriano Aguzzi

DOI: [10.15252/emmm.202012739](https://doi.org/10.15252/emmm.202012739)

Corresponding authors: Adriano Aguzzi (adriano.aguzzi@usz.ch) , Simone Hornemann (simone.hornemann@usz.ch)

Review Timeline:	Transfer from Review Commons:	14th May 20
	Editorial Decision:	10th Jun 20
	Revision Received:	10th Jul 20
	Accepted:	10th Jul 20

Editor: Celine Carret

Transaction Report:

(Note: This manuscript was transferred to EMBO Molecular Medicine following peer review at Review Commons. With the exception of the correction of typographical or spelling errors that could be a source of ambiguity, letters and reports are not edited. Depending on transfer agreements, referee reports obtained elsewhere may or may not be included in this compilation. Referee reports are anonymous unless the Referee chooses to sign their reports.)

Review #1

1. How much time do you estimate the authors will need to complete the suggested revisions:

Estimated time to Complete Revisions (Required) (Decision Recommendation)

Less than 1 month

2. Evidence, reproducibility and clarity:

Evidence, reproducibility and clarity (Required)

This is a very interesting article with important implications in the prion field. It is extremely well detailed and exquisitely well written. The objective of the article is very clear and the results obtained are not only interesting but also have very important implications for understanding prion diseases. I have some comments and a few minor concerns. Abstract: Although it is clear and direct, the last sentence where it refers to "a link to the low incidence of spontaneous prion diseases in human populations", is not easy to understand without a detailed explanation as given in the Discussion. I suggest a re-wording. Results: It is clear how these Fabs act in preventing prion-induced neurotoxicity as shown in the COCS model. In addition to this effect, they also inhibit prion spreading, although this appears to be a lesser effect than inhibition of neurotoxicity. Thus, it would be interesting to discuss the possible effect of a Fab therapy, which provide a fully inhibition of the neurotoxicity but only partially inhibition of the prion propagation. The therapeutic effect of the Fabs in the cell model was performed by adding the Fabs to the medium 1 h after infection and during splitting. Is there any study that evaluates the effect of Fabs added to the medium before inoculation or at later times? Discussion: The authors repeatedly refer to the toxicity that antibodies against GD might have. Related to this, there is currently a therapy (experimental medicine) in humans using an antibody against this region. Perhaps it would be interesting to make a comment on this. Page 15. I have found the speculative comment: "Accordingly, clinically silent prion generation may occasionally occur in healthy individuals. PrPSc aggregates arising de novo may result in exposure of neoepitopes and/or epitopes occluded in cell-borne PrPC." interesting. However, some of the auto-antibodies found in healthy humans are against a region believed to be structurally unaltered in PrPSc, which it doesn't fit with the theory of exposure to neo-epitopes.

3. Significance:

Significance (Required)

The advance is highly significance for two reasons: 1) the tools that the authors have generated are really useful for the community and 2) The fact the healthy humans can generate anti-PrP antibodies is completely new and open new ways to understand the prion diseases mechanisms. The audience is principally for those working on prion and prion-like diseases. My expertise is in prion and prion-like diseases.

Review #2

1. How much time do you estimate the authors will need to complete the suggested revisions:

Estimated time to Complete Revisions (Required)

(Decision Recommendation)

Less than 1 month

2. Evidence, reproducibility and clarity:

Evidence, reproducibility and clarity (Required)

****Summary:**** The authors extensively and rigorously characterized a subset of antibodies to PrP identified in a human Fab phage display library. These selected antibodies were compared and found to be similar to repertoires of naturally occurring human antibodies present in circulating B cells. Profiling of antibodies harvested from an unbiased 38,000 patient population uncovered the presence of high titer anti-PrP autoantibodies in 21 individuals sharing no specific pathologies. This finding demonstrates the presence of apparently innocuous immunity to prion in an unselected population. Based also on "the reported lack of such antibodies in carriers of disease-associated PRNP mutations" the authors propose that the low incidence of "spontaneous" prion diseases may be linked to the presence of these protective antibodies in the general population. ****Major comments:**** This is a technically advanced and carefully executed study that clearly demonstrate the presence of natural autoantibodies to PrP, some of which show protective properties, in an unselected human population. Although this finding is interesting on its own right, its impact on issues such as incidence of sporadic prion diseases is unclear given that apparently only 0.06% of the nearly 38,000 subjects examined carried these antibodies "in high titer". Furthermore, this reviewer could not locate the base of the pivotal statement made in the Abstract that these autoantibodies lack in carriers of disease-associated PRNP mutations. These two points need to be clarified. The manuscript suffers for the excessive amount of data that are crammed in the five figures. Combined these figures display a total of 33 panels some of which are quite complicated. The authors should be more selective and roll over some of the nonessential information i.e. that related to

methodology, to the Supplement. ****Minor comments:**** The use of acronyms is excessive and should be reduced (see for example COCS). The legends need to be carefully checked for clarity, especially figure 4

3. Significance:

Significance (Required)

Significance See above Referees Cross Commenting I agree with most of the comments by Reviewers 1 and 3. However, my queries remain.

Review #3

1. How much time do you estimate the authors will need to complete the suggested revisions:

Estimated time to Complete Revisions (Required)

(Decision Recommendation)

Less than 1 month

2. Evidence, reproducibility and clarity:

Evidence, reproducibility and clarity (Required)

The manuscript by Senatore et al. is large scale study looking for natural human antibodies directed against prion protein (PrP). Using a synthetic human Fab phage display library, they found and characterized multiple human anti-PrP Fabs most of which recognized epitopes to a region of PrP from amino acid residues 92-120. Based on this information, they searched for and found low affinity, long-lived anti-PrP antibodies in both a repertoire of human antibodies and in 27 of almost 38,000 human clinical samples. They speculate that anti-PrP antibodies may help to protect against sporadic forms of prion disease and conclude that they may represent a source of potential immunotherapeutics against human prion infection. ****Minor comments:**** 1) On page 10, the authors state that Fab71 (Figure 3e) and Fab100 (Extended data Figure 7) substantially lowered PrPSc levels in prion-infected cells. However, in both cases, only about half of the cultures tested showed less PrPSc than either the control samples or samples treated with other Fabs. This variability undercuts the conclusion that what they are observing is a substantial, reproducible effect. The authors should consider moderating their conclusion somewhat to better fit the data. 2) In figure 2, the legend to panel a does not match the figure. Fab3 and Fab71 are represented by the blue lines, not the red lines as stated in the legend. 3) In the legend to Extended data figure S4, please

give the epitopes to Fab10 and Fab53. 4) In Figure 3c, the lines indicating the significant groups are not well-aligned. In the left side of the panel, the lines should connect the dark gray control group squares with the Fab25 pink diamonds. Likewise, in the right side of the panel, the lower set of lines should connect the dark gray control group squares with the Fab83 dark blue triangles.

3. Significance:

Significance (Required)

This is an extensive, well-written study which provides significant data suggesting that humans can make anti-PrP antibodies. This is a novel finding that raises important questions about how the body may respond to spontaneous formation of infectious prions. Technically, the study is sound with appropriately interpreted data. Overall the study and the antibodies it characterizes, some of which are novel, will be of interest to prion researchers. Referees cross commenting I agree with the comments of both reviewers. The suggestion of reviewer #2 to move methodology-related panels in the main figures to supplemental data would make it much easier for the reader to focus on the critical experimental data.

Authors' Response to Referees in Review Commons

Reviewer #1 (Evidence, reproducibility and clarity (Required)):

Comment 1: Abstract: Although it is clear and direct, the last sentence where it refers to "a link to the low incidence of spontaneous prion diseases in human populations", is not easy to understand without a detailed explanation as given in the Discussion. I suggest a re-wording.

Response 1: We have reworded the sentence in the abstract and given more explanation in the discussion (see also Reviewer 2, Comment 2).

Comment 2: Results: It is clear how these Fabs act in preventing prion-induced neurotoxicity as shown in the COCS model. In addition to this effect, they also inhibit prion spreading, although this appears to be a lesser effect than inhibition of neurotoxicity. Thus, it would be interesting to discuss the possible effect of a Fab therapy, which provide a fully inhibition of the neurotoxicity but only partially inhibition of the prion propagation.

Response 2: As suggested by the reviewer, we have added appropriate text to the discussion to comment on the option of a potential Fab therapy with a fully inhibition of neurotoxicity and partially inhibition of prion propagation.

Comment 3: The therapeutic effect of the Fabs in the cell model was performed by adding the Fabs to the medium 1 h after infection and during splitting. Is there any study that evaluates the effect of Fabs added to the medium before inoculation or at later times?

Response 3: The goal of these experiments was to investigate whether the antibodies in question would counteract prion infections in principle, rather than performing a precise range-finding of the optimal therapeutic window. We have opted to not add the Fabs before inoculation, because past experience (and many papers) show that the "prophylactic" treatment rarely correlated with post-exposure efficacy. We also have not treated the cells after prion infection at later time points, because the data at later time points may be less pronounced and more variable.

As for the treatment of cells with anti-PrP antibodies prior to exposure to prions, a study has been conducted in N2a cells (Pankiewicz J et al., 2006, <https://www.ncbi.nlm.nih.gov/pmc/articles/PMC1779824/>). There, preincubation of N2a cells with mouse monoclonal anti-PrP antibodies (Mabs) before prion infection (22L) and preincubation of the inoculum with Mabs before infection of the cells led to a significant reduction in PrP^{Sc} levels as assessed by proteinase-K Western blot. This paper is now discussed in our manuscript.

Comment 4: Discussion: The authors repeatedly refer to the toxicity that antibodies against GD might have. Related to this, there is currently a therapy (experimental medicine) in humans using an antibody against this region. Perhaps it would be interesting to make a comment on this.

Response 4: Our findings (Sonati et al, Nature 2013, and several following papers) are fundamentally incompatible with those of the London lab on the toxicity of anti-GD antibodies, and elsewhere I have warned loudly against the use of such antibodies in humans. However, this discussion is peripheral to the findings presented here. We have added some text to the discussion but we would rather not expand on this specific issue.

Comment 5: Page 15. I have found the speculative comment: "Accordingly, clinically silent prion generation may occasionally occur in healthy individuals. PrP^{Sc} aggregates arising de novo may result in exposure of neoepitopes and/or epitopes occluded in cell-borne PrP^C." interesting. However, some of the auto-antibodies found in healthy humans are against a region believed to be structurally unaltered in PrP^{Sc}, which it doesn't fit with the theory of exposure to neo-epitopes.

Response 5: I still believe that my hypothesis is viable, but of course I concede that – thus far – I have no supporting data. We have therefore modified the text to alleviate this comment.

Reviewer #2:

Comment 1: This is a technically advanced and carefully executed study that clearly demonstrate the presence of natural autoantibodies to PrP, some of which show protective properties, in an unselected human population. Although this finding is interesting on its own right, its impact on issues such as incidence of sporadic prion diseases is unclear given that apparently only 0.06% of the nearly 38,000 subjects examined carried these antibodies "in high titer".

Response 1: We agree with the reviewer and have modified the statement as follows: *"The frequency of high-titer anti-PrP antibody carriers (0.06%) is much lower than the occurrence of Fab71-like HCDR3 sequences in published human repertoires. This discrepancy could mean that most anti-PrP specificities exist in a dormant state, or are expressed as B-cell receptors, but do not produce circulating antibodies. It will be interesting to discover the triggers that may ignite antibody production and, possibly, afford protection against prions"*. The discrepancy between the frequency of anti-PrP antibodies found in the plasma screen and by analysis of the antibody repertoires in the NGS datasets could stem from the fact that most anti-PrP specificities exist in a dormant state, or are expressed as B-cell receptors, but do not produce circulating antibodies (Joseena Iype et al., J Immunol 2019; now also included in the manuscript).

Comment 2: Furthermore, this reviewer could not locate the base of the pivotal statement made in the Abstract that these autoantibodies lack in carriers of disease-associated PRNP mutations. These two points need to be clarified.

Response 2: The statement refers to the study by Frontzek et al. (citation #48: Frontzek, K. *et al. Autoantibodies against the prion protein in individuals with PRNP mutations* Neurology <https://n.neurology.org/content/early/2020/02/25/WNL.0000000000009183?rss=1>)

Although listed in the references, the citation got lost in the discussion. We have inserted the reference again.

Comment 3: The manuscript suffers for the excessive amount of data that are crammed in the five figures. Combined these figures display a total of 33 panels some of which are quite complicated. The authors should be more selective and roll over some of the nonessential information i.e. that related to methodology, to the Supplement.

Response 3: We agree with the reviewer and have moved several panels to the Supplement.

Comment 4: The use of acronyms is excessive and should be reduced (see for example COCS).

Response 4: We have attempted to reduce the number of acronyms. We have however introduced the term COCS in Falsig et al., Nature Neuroscience 2007, and have used it regularly in more than a dozen follow-up papers.

Comment 5: The legends need to be carefully checked for clarity, especially figure 4

Response 5: We have revised the legends to improve their clarity.

Reviewer #3:

Comment 1: On page 10, the authors state that Fab71 (Figure 3e) and Fab100 (Extended data Figure 7) substantially lowered PrP^{Sc} levels in prion-infected cells. However, in both cases, only about half

of the cultures tested showed less PrP^{Sc} than either the control samples or samples treated with other Fabs. This variability undercuts the conclusion that what they are observing is a substantial, reproducible effect. The authors should consider moderating their conclusion somewhat to better fit the data.

Response 1: We agree with the reviewer. The effect of Fab71 and Fab100 in reducing PrP^{Sc} levels in cells as compared to control samples and samples treated with the other Fabs is only partially present and variable among the replicates, but still statistically significant (One-way ANOVA; $p < 0.05$ for Fab100 and $p < 0.01$ for Fab71). We have moderated our conclusion accordingly.

Comment 2: In figure 2, the legend to panel a does not match the figure. Fab3 and Fab71 are represented by the blue lines, not the red lines as stated in the legend.

Response 2: We thank the referee for pointing this out. We have now corrected it.

Comment 3: In the legend to Extended data figure S4, please give the epitopes to Fab10 and Fab53.

Response 3: We have included the epitopes of these two Fabs (OR₅₁₋₉₁ for Fab10 and CC2-HC₉₂₋₁₂₀ for Fab53) in the Figure legend.

Comment 4: In Figure 3c, the lines indicating the significant groups are not well-aligned. In the left side of the panel, the lines should connect the dark gray control group squares with the Fab25 pink diamonds. Likewise, in the right side of the panel, the lower set of lines should connect the dark gray control group squares with the Fab83 dark blue triangles.

Response 4: We have corrected this issue.

Comment 5: I agree with the comments of both reviewers. The suggestion of reviewer #2 to move methodology-related panels in the main figures to supplemental data would make it much easier for the reader to focus on the critical experimental data.

Response 5: See response to comment 2 of reviewer 2.

With all issues addressed, we hope that our revised manuscript will now be found suitable for proceeding to the next steps.

10th Jun 2020

Dear Prof. Aguzzi,

Thank you for your submission to EMBO Molecular Medicine.

Please find below the two sets of comments I have now received regarding your revised article initially reviewed at Review Commons. In addition to getting back to two referees, I had sought advice on your article regarding its suitability for EMBO Molecular Medicine and our advisor shared the referees support. Reviewers and advisor are all positive about the article timeliness and suitability for publication. Still, a few minor items are needed and I would like to ask you to perform the following:

1) Please provide a point-by-point letter INCLUDING my comments as well as the advisor's report and your detailed responses to their comments (as Word file).

2) Please carefully check the authors guidelines for formatting your supplemental information:

Expanded view and Appendix (see:

<https://www.embopress.org/page/journal/17574684/authorguide#expandedview>).

Supplementary information should be split into Expanded View figures (max of 5) - labelled and called out as Fig. EV1-5 (legends in main article) - and Appendix document provided as pdf, including a Table of content of the 1st page, and Figures Sx and Tables Sy all together.

As such we'd like to propose that you relabel and change the call outs for Table S1 -> Appendix Table S1, Table S2 -> Appendix Table S2, Table S3 -> Dataset EV1 (because it's provided as an excel document, please add the legend on top of the file or as a separate sheet within the excel document), Table S4 -> Appendix Table S3, Table S5 -> Appendix Table S4, Table S6 -> Appendix Table S5.

3) Figures and EV figures must be provided as individual separate files of high resolution. The legends must be within the main article, at the end.

4) the main article must be provided as a word file

5) In the main manuscript file, please do the following:

- correct/answer the track changes suggested by our data editors by working from the attached/uploaded document
- add up to 5 keywords
- references are not numbered but alphabetical and listed at the end with 10 authors followed by et al.
- in M&M, provide all antibody dilutions that were used for each antibody
- in M&M, the statistical paragraph should reflect all information that you have filled in the Authors checklist (see below), especially regarding randomisation, blinding, replication.
- indicate in legends exact n= and exact p= values, not a range, along with the statistical test used. Some people found that to keep the figures clear, providing an Appendix table Sx with all exact p-values was preferable. You are welcome to do this if you want to.
- in M&M, for animal work, gender, age, origin of the animals and genetic background must be indicated, along with housing conditions.

- in M&M, include a statement that informed consent was obtained from all human subjects.
- competing interest should be relabelled as Conflict of Interest

6) Source Data:

We encourage the publication of source data, particularly for electrophoretic gels, blots, but also microscopy images with the aim of making primary data more accessible and transparent to the reader. Would you be willing to provide a PDF file per figure that contains the original, uncropped and unprocessed scans of all or key gels used in the figure (including molecular weight markers)? The PDF files should be labeled with the appropriate figure/panel number (1 file/figure), and should have molecular weight markers; further annotation may be useful but is not essential. The PDF files will be published online with the article as supplementary "Source Data" files. If you have any questions regarding this just contact me.

7) EMBO Molecular Medicine now requires a complete author checklist (<http://embomolmed.embopress.org/authorguide#editorial3>) to be submitted with all revised manuscripts. Please use the checklist as guideline for the sort of information we need WITHIN the manuscript. This is particularly important for animal reporting, the use of human samples, antibody dilutions and exact p- and n-values that should be indicated instead of a range, along with the right justified statistical test.

This file will be published along the Review Process File.

8) For more information: There is space at the end of each article to list relevant web links for further consultation by our readers. Could you identify some relevant ones and provide such information as well? Some examples are patient associations, relevant databases, OMIM/proteins/genes links, author's websites, etc...

9) The Paper Explained: EMBO Molecular Medicine articles are accompanied by a summary of the articles to emphasize the major findings in the paper and their medical implications for the non-specialist reader. Please provide a draft summary of your article highlighting

- the medical issue you are addressing, = Problem
- the results obtained = Results
- their clinical impact = Impact

10) Every published paper now includes a 'Synopsis' to further enhance discoverability. Synopses are displayed on the journal webpage and are freely accessible to all readers. They include a short stand first (maximum of 300 characters, including space) as well as 2-5 one sentence bullet points that summarise the paper. Please write the bullet points to summarise the key NEW findings. They should be designed to be complementary to the abstract - i.e. not repeat the same text. We encourage inclusion of key acronyms and quantitative information (maximum of 30 words / bullet point). Please use the passive voice. Please attach these in a separate file or send them by email, we will incorporate them accordingly.

You are also encouraged to suggest a striking image or visual abstract to illustrate your article. If you do please provide a jpeg file 550 px-wide x (250-400)-px high.

11) As part of the EMBO Publications transparent editorial process initiative (see our Editorial at

<http://embomolmed.embopress.org/content/2/9/329>), EMBO Molecular Medicine will publish online a Review Process File (RPF) to accompany accepted manuscripts.

In the event of acceptance, this file will be published in conjunction with your paper and will include the anonymous referee reports, your point-by-point response and all pertinent correspondence relating to the manuscript. Let us know whether you agree with the publication of the RPF.

12) Please note that we now mandate that all corresponding authors list an ORCID digital identifier. This takes less than 90 seconds to complete. We encourage all authors to supply an ORCID identifier, which will be linked to their name for unambiguous name identification. Your co-corresponding author Dr. Hornemann must entrain ORCID number.

13) Data and software availability:

To list the primary data generated in your study, we would kindly ask you to include a formal "Data and software availability" section (after Materials & Methods) that follows the example below:

- [data type]: [full name of the resource] [accession number/identifier] ([doi or URL or identifiers.org/DATABASE:ACCESSION])

example:

* RNA-Seq data: Gene Expression Omnibus GSExxxxx
(<https://www.ncbi.nlm.nih.gov/geo/query/acc.cgi?acc=GSExxxxx>)

You can submit your revised files by logging onto our online manuscript tracking system or simply follow this link:

Link Not Available

I hope that the referees' comments do not prove too problematic to address and I look forward to reading your next version.

Yours sincerely,

Celine Carret

Celine Carret, PhD
Senior Editor
EMBO Molecular Medicine

*** IMPORTANT INFORMATION ***

When submitting your revised manuscript , please include:

- 1) a .doc formatted version of the manuscript text (including Figure legends and tables)
- 2) Separate figure files
- 3) a letter INCLUDING the reviewer's reports and your detailed responses to their comments.

Thank you,

Celine Carret

Celine Carret, PhD
Senior Editor
EMBO Molecular Medicine

***** Reviewer's comments *****

Referee #1 (Remarks for Author):

The authors have addressed the issues I raised in an acceptable way.

Referee #2 (Remarks for Author):

In the revised manuscript, the authors have satisfactorily addressed all of the comments in my original review.

Advisor:

... overstatements such as: The existence of protective anti-prion antibodies in unbiased human immunological repertoires, combined with the reported lack of such antibodies in carriers of disease-associated PRNP mutations, suggests a link to the low incidence of spontaneous prion diseases in human populations.

How do we know that these antibodies are indeed protective in humans? After all, some of the antibodies are against regions, which are structurally unaltered as also pointed out by one of the reviewers.

[Can] they explain their screens and findings in more detail, as suggested by one reviewer (see comment 3 of reviewer 2)? I think the paper deserves it. I don't like that some of the data are now buried in the supplements, that does not increase readability!

**** Reviewer's comments ****

Referee #1 (Remarks for Author):

The authors have addressed the issues I raised in an acceptable way.

Referee #2 (Remarks for Author):

In the revised manuscript, the authors have satisfactorily addressed all of the comments in my original review.

Advisor:

... overstatements such as: The existence of protective anti-prion antibodies in unbiased human immunological repertoires, combined with the reported lack of such antibodies in carriers of disease-associated PRNP mutations, suggests a link to the low incidence of spontaneous prion diseases in human populations.

How do we know that these antibodies are indeed protective in humans? After all, some of the antibodies are against regions, which are structurally unaltered as also pointed out by one of the reviewers.

Response: We agree with the advisor that our data did not specifically show that anti-PrP antibodies from human plasma are indeed protective in humans. We therefore toned down this statement by eliminating the term "protective" and adjusting the next sentence appropriately.

[Can] they explain their screens and findings in more detail, as suggested by one reviewer (see comment 3 of reviewer 2)? I think the paper deserves it. I don't like that some of the data are now buried in the supplements, that does not increase readability!

Response: We have moved back several data related to the antibody screen in human plasma to main Figure 5, including the schematic description of the workflow of the screen (panel A), the demographic data of the hits (panel d) and the longitudinal analysis confirming the high titer of anti- PrP antibodies for some hits.

10th Jul 2020

Dear Prof. Aguzzi,

We are pleased to inform you that your manuscript is accepted for publication and will be sent to our publisher to be included in the next available issue of EMBO Molecular Medicine.

If you want to receive an e-mail alert regarding its publication as well as other EMBO Mol Med content, register here: <http://embomolmed.embopress.org/alerts>

Our RSS feeds can be found at feed:
<http://embomolmed.embopress.org/rss>

Please read below for additional IMPORTANT information regarding your article, its publication and the production process.

Congratulations on your interesting work and thank you for choosing EMBO Molecular Medicine,

Celine

Celine Carret, PhD
Senior Editor
EMBO Molecular Medicine

Corresponding Author Name: Adriano Aguzzi, Simone Hornemann

Manuscript Number: EMM-2020-12739 | [RC-2020-00194]